# Demystifying LLM-as-a-Judge: Analytically Tractable Model for Inference-Time Scaling

**Indranil Halder** [1]   **Cengiz Pehlevan** [1 2 3]

## Abstract

Recent developments in large language models have shown advantages in reallocating a notable share of computational resource from training time to inference time. However, the principles behind inference time scaling are not well understood. In this paper, we introduce an analytically tractable model of inference-time scaling: Bayesian linear regression with a reward-weighted sampler, where the reward is determined from a linear model, modeling LLM-as-a-judge scenario. We study this problem in the high-dimensional regime, where the deterministic equivalents dictate a closed-form expression for the posterior predictive mean and variance. We analyze the generalization error when training data are sampled from a teacher model. We draw $k$ inference-time samples and select via softmax at a temperature applied to a quadratic reward. When the reward is not too different from the teacher, the generalization error decreases monotonically with increasing inference time samples $k$. However, the specific reward that optimizes inference-time selection generally differs from the teacher. In contrast, substantial reward misspecification induces a finite optimal $k$ beyond which more sampling can increase the generalization error. For fixed $k$, there exists an optimal sampling temperature. We experimentally verify these facts in large language model inference with an additional large language model as a judge. In the "best-of-$k$" limit with the teacher as reward, we theoretically show that the generalization error decays as $\Theta(1/k^2)$ and determine the leading coefficient via extreme value theory. These formulas

delineate domains where scaling inference-time computation is provably preferable to collecting more data. Finally, we demonstrate that when task difficulty increases, the previously mentioned advantage of inference-time compute degrades.

## 1. Introduction

Scaling training compute via larger models and more data drives dramatic gains (Hestness et al., 2017; Kaplan et al., 2020; Hoffmann et al., 2022). In parallel, across tasks, allowing models to 'think longer' at inference by sampling multiple candidates, re-ranking with a reward, or aggregating votes consistently improves precision and reliability (Wang et al., 2023; Zheng et al., 2023; Zhang et al., 2024; Wu et al., 2024; Snell et al., 2025). Best-of-$k$ (choose the highest-reward sample based on the feedback of a judge large language model) and majority voting (choose the consensus among the generated answers) have become standard inference-time tools (Brown et al., 2024; Schaeffer et al., 2025a; Chen et al., 2024a; Huang et al., 2025a) along with self-verification (Saunders et al., 2022; Weng et al., 2023).

Despite widespread adoption, key questions for inference-time computation lack crisp answers. How to optimally configure inference-time sampling to minimize generalization error under realistic compute constraints and how to allocate compute optimally between pretraining and inference time remain open questions (Wu et al., 2024). Which reward model should we use for inference? What is the appropriate inference-time sampler, e.g., temperature settings? How large should $k$ be and when do more samples stop helping? How should we allocate a fixed compute budget between training and inference to minimize generalization error? We lack a simple, solvable model that gives us intuitions and actionable prescriptions.

To fill this gap, we propose a minimal and analytically tractable setting: Bayesian regression based on a teacher-student setting with a controlled reward and a temperature-dependent inference-time sampler, in which best-of-$k$ appears as limit. This setup is a proxy for LLM-as-a-judge providing reward for the generated answers from a base-LLM. We theoretically study the generalization error $\delta$ as a

---

[1]John A. Paulson School of Engineering And Applied Sciences, Harvard University [2]Kempner Institute for the Study of Natural and Artificial Intelligence, Harvard University [3]Center for Brain Science, Harvard University. Correspondence to: Indranil Halder <ihalder@g.harvard.edu>.

*Proceedings of the 43$^{rd}$ International Conference on Machine Learning*, Seoul, South Korea. PMLR 306, 2026. Copyright 2026 by the author(s).

function of the size of the training data set $n$, the dimension of the data $d$, the number of inference time samples $k$, the sampling temperature $T$ and the reward parameter $\mathbf{w}_R$ and demonstrate various optimality conditions on these parameters. We generalize these theoretical discussion to generic probabilistic models in Appendix F.

This simple model recapitulates existing findings on large language model (LLM) inference and yields novel predictions that we evaluate in this paper. It reproduces the empirical observation that unbounded increases in the number of inference-time samples do not confer additional benefit (Snell et al., 2025). Furthermore, it predicts the existence of an optimal temperature for the reward process. Further, strong rewards shift optimal temperature to a lower value. We empirically examine and confirm this prediction using Llama-3-8B-Instruct, DeepSeek-R1-Distill-Qwen-1.5B etc.

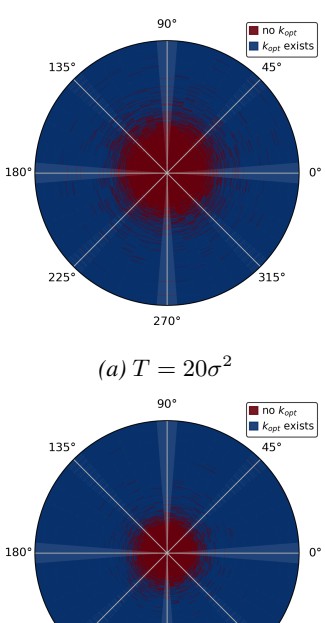

*(a) $T = 20\sigma^2$*

*(b) $T = 10\sigma^2$*

*Figure 1.* In the plot the radial distance is the magnitude $c$ of the vector $\mathbf{w}_R - \mathbf{w}_T$ and the polar variable is the angle $\theta$ between $\mathbf{w}_R - \mathbf{w}_T$ and $\mathbf{w}_T$. We have chosen $S = 1, \sigma = 10^{-4}, \gamma = 10^{-3}, d = 2, n = 10^4$ and sampled teacher weight $\mathbf{w}_T = (c\cos\theta_T, c\sin\theta_T) \sim \mathcal{N}(0, 2^2\mathbf{I})$. We have parameterized the reward weight as follows: $\mathbf{w}_R = \mathbf{w}_T + (c\cos(\theta_T + \theta), c\sin(\theta_T + \theta)), \theta \in [0, 2\pi)$. See section 2 for details of notation and conventions. From the plot, we see that as temperature $T$ decreases the domain, where generalization error $\delta$ decreases monotonically with the increase in inference-time samples $k$, shrinks. Similar result is presented in remark 2 for the proportional limit.

At the technical level our contributions are the following:

- We propose a solvable model for inference-time scaling: Bayesian regression where the ground truth is given by the teacher model $y = \mathbf{w}_T \cdot \mathbf{x}/\sqrt{d}$ and the reward function is quadratic $r(y, \mathbf{x}) = -(y - \mathbf{w}_R \cdot \mathbf{x}/\sqrt{d})^2$. We generate $k$ samples at inference and choose using a softmax at temperature $T$ over the reward. This method is conceptually close to (Aminian et al., 2025; Faria and Smith, 2025). We present a formula for the generalization error $\delta$ in the proportional limit: $d \to \infty, n \to \infty$ with $\alpha = d/n$ fixed.

- In order to derive analytic insights about the inference time optimization, we derive a series expansion for $\delta$ around large $T$, making explicit its dependence on $n$, $k$, and the alignment between $\mathbf{w}_R$ and $\mathbf{w}_T$. The analytical expansion shows a sharp dependence on reward quality: when $\mathbf{w}_R$ is sufficiently close to $\mathbf{w}_T$, i.e., small $\|\mathbf{w}_R - \mathbf{w}_T\|/\|\mathbf{w}_T\|$, increasing $k$ monotonically decreases $\delta$ (see Figure 1), and the reward $\mathbf{w}_R$ that optimizes inference-time selection generally *differs* from the data-generating teacher $\mathbf{w}_T$ (Remark 2, Figure 3). In contrast, when $\mathbf{w}_R$ is poorly aligned, $\delta$ is non-monotone in $k$, yielding an *optimal* finite $k$ (Remark 3, Figure 4, 7a), echoing phenomena observed empirically in large language models (Snell et al., 2025). Furthermore we show that at fixed $k$, there exists an *optimal* temperature $T$ for the rewarding process (Remark 4, Figure 5, 7b). A similar yet distinct observation has been made in large language models (Du et al., 2025) for sampling temperature of the model itself. We emphasize that our result is about the sampling temperature of the rewarding process - different from the sampling temperature of the language model. In Appendix F, we discussion statistical origin of these results for a generic probabilistic model with a general reward.

- To explore the best-of-k case, we consider the $T = 0$ limit. Using extreme value theory, we analytically prove that the expectation value of $\delta$ scales as $\Theta(k^{-2})$ at large $k$ when we have access to the teacher, i.e, $\mathbf{w}_R = \mathbf{w}_T$ (Remark 5, Figure 6, 9). In Appendix F, we show that in general depending on the coverage coefficient $\beta$ of the predictive distribution near the target one can get more general scaling law $\Theta(k^{-\frac{2}{1+\beta}})$ that fits the plot Figure 9 in large language models. Further, based on our theoretical analysis in best-of-$k$ limit, we quantify the parametric region where scaling inference-time compute is more beneficial compared to training compute. Finally we note that when task difficulty increases, the previously mentioned advantage of inference-time compute degrades.

- To test our predictions in realistic settings, we have generated inference time samples from Llama-3-8B-

Instruct and DeepSeek-R1-Distill-Qwen-1.5B on openai/gsm8k validation dataset (see Appendix H.2 for the details and more empirical results). For each question and response pair we used Mistral-7B-Instruct-v0.3 or Qwen2.5-Math-PRM-7B to generate a reward score. Finally, we observed the existence of an optimal $k, T$ similar to the theoretical predictions above (see Figure 7 and 8). In particular for the best-of-$k$ strategy, Figure 9 shows scaling laws similar to the one observed in theoretical analysis for a nearly ideal reward model.

Now we turn to survey the ideas in the literature related to our work.

### Related Works

**Method of deterministic equivalence.** In the context of linear regression (Krogh and Hertz, 1992; Dicker, 2016; Dobriban and Wager, 2018; Nakkiran, 2019; Advani et al., 2020; Hastie et al., 2022), kernel regression (Sollich, 1998; Sollich and Halees, 2002; Bordelon et al., 2020; Canatar et al., 2021; Spigler et al., 2020; Simon et al., 2021; Loureiro et al., 2021), and random feature models (Hastie et al., 2022; Louart et al., 2018; Mei and Montanari, 2022; Adlam and Pennington, 2020; d'Ascoli et al., 2020; d'Ascoli et al., 2020; Loureiro et al., 2021; Bahri et al., 2022; Zavatone-Veth and Pehlevan, 2023a; Dhifallah and Lu, 2020; Hu and Lu, 2022; Maloney et al., 2022; Bach, 2024) method of deterministic equivalence (Voiculescu et al., 1992; Zee, 1996) has been used extensively for discussions of higher dimensional statistics (Misiakiewicz and Saeed, 2024; Atanasov et al., 2024). These ideas have been used to discuss training time scaling laws in simple models (Spigler et al., 2020; Bordelon et al., 2020; Bahri et al., 2022; Maloney et al., 2022; Simon et al., 2021; Bordelon et al., 2024; Zavatone-Veth and Pehlevan, 2023b; Paquette et al., 2024; Lin et al., 2024; Bordelon et al., 2025). We use this technique to simplify the posterior probability distribution of the Bayesian regression model.

**Inference-time scaling.** A growing body of work investigates how to allocate and exploit inference-time compute to improve predictive performance, with empirical gains reported across tasks and domains based on majority voting (Chen et al., 2024a; Snell et al., 2025; Setlur et al., 2025; Arora and Zanette, 2025; Wu et al., 2024; Liu et al., 2025; Du et al., 2025; Huang et al., 2025a) or a best-of-$k$ strategy (Wang et al., 2023; Yao et al., 2023a; Brown et al., 2024; Levi, 2024; Schaeffer et al., 2025b; Huang et al., 2025b; Chen et al., 2024b; Du et al., 2025; Chen et al., 2025; Aminian et al., 2025; Faria and Smith, 2025; Khalaf et al., 2026). These procedures are often paired with reasoning-oriented prompting and structured search that expand the candidate set before selection (Wei et al., 2022; Yao et al.,

2023b; Wu et al., 2024). Recent work has also begun to compare different inference-time scaling paradigms theoretically. For majority-vote-style compound systems, Chen et al. (2024a) develop a theoretical model for classification problems and analyze when repeated calls to a language model improve accuracy. Relatedly, Huang et al. (2025c) provide a theoretical account of self-consistency-based selection and compare its sample complexity with other test-time scaling procedures. In another direction, Di et al. (2025) propose a method that combines ideas from majority voting and best-of-$k$: the algorithm first uses frequency information to restrict attention to plausible candidate answers and then applies reward-based selection within this filtered set. This perspective highlights an important limitation of pure best-of-$k$ selection: when the reward model is imperfect, aggressively selecting the highest-reward sample can amplify reward misspecification. The use of consensus or frequency information can therefore act as a stabilizing mechanism. More close to our work is the scaling law of (Brown et al., 2024; Schaeffer et al., 2025b) for best-of-$k$ strategy. For a given trained model, both these works explain pass@k performance for correctness of the prediction. We study similar questions with imperfect reward and discuss best-of-$k$ (and more generally softmax weighted selection) generalization error when reasoning is taken into account both theoretically and empirically.

## 2. Problem Setup: Bayesian Regression With Reward-weighted Sampling

We start by introducing our solvable model.

### 2.1. Training Method - Prior and Posterior Distribution

We study a supervised regression setting with a linear teacher model that maps inputs to outputs and then adds observation noise. Throughout, let $\mathbf{x} \in \mathbb{R}^d$ denote an input vector drawn from a zero-mean Gaussian with covariance $\mathbf{\Sigma}$, written $\mathbf{x} \sim \mathcal{N}(0, \mathbf{\Sigma})$. We assume $\mathrm{Tr}(\mathbf{\Sigma}) = \Theta_d(d)$ so that the total feature variance scales linearly with dimension; a canonical case is $\mathbf{\Sigma} = \mathbf{I}$. The teacher parameter $\mathbf{w}_T$ is taken to have norm $\|\mathbf{w}_T\|^2 = d$, and the output is given by:

$$y = \mathbf{w}_T \cdot \frac{\mathbf{x}}{\sqrt{d}} + \eta, \quad \eta \sim \mathcal{N}(0, \sigma^2). \quad (1)$$

Given a training set $\mathcal{D} = \left\{ (\mathbf{x}^i, y^i)_{i=1}^n \right\}$ sampled i.i.d. from the teacher, we adopt a Bayesian linear regression perspective with an isotropic Gaussian prior on the weights, $\mathcal{N}(0, \gamma^2 \mathbf{I})$. Bayes' rule yields the posterior distribution over weights:

$$p(\mathbf{w}|\mathcal{D}) = \frac{p(\mathcal{D}|\mathbf{w})p(\mathbf{w})}{p(\mathcal{D})}. \quad (2)$$

Predictions for a new test input $\mathbf{x}$ are obtained by marginalizing the likelihood under this posterior, producing the *posterior predictive* distribution:

$$p(y|\mathbf{x}, \mathcal{D}) = \int d\mathbf{w}\, p(\mathbf{w}|\mathcal{D})p(y|\mathbf{x}, \mathbf{w}). \qquad (3)$$

Next we state the standard result that makes predictive distribution explicit.

*Remark* 1. Analytical formula for the posterior predictive is given by

$$p(y|\mathbf{x}, \mathcal{D}) = \mathcal{N}\left(\boldsymbol{\mu}\cdot\frac{\mathbf{x}}{\sqrt{d}}, \frac{\mathbf{x}}{\sqrt{d}}^{\top}\boldsymbol{\Omega}\frac{\mathbf{x}}{\sqrt{d}} + \sigma^2\right) \qquad (4)$$

$$\boldsymbol{\mu} = \frac{1}{\sigma^2}\boldsymbol{\Omega}\sum_{i=1}^{n} y^i \frac{\mathbf{x}^i}{\sqrt{d}}, \quad \boldsymbol{\Omega}^{-1} = \frac{1}{\sigma^2}\sum_{i=1}^{n}\frac{\mathbf{x}^i}{\sqrt{d}}\frac{\mathbf{x}^i}{\sqrt{d}}^{\top} + \frac{1}{\gamma^2}\mathbf{I} \qquad (5)$$

*Proof-sketch.* This is a standard result (Bishop, 2013).

### 2.2. Inference-time Sampling And The Reward Model

Suppose that we have a reward model that evaluates our predictions, $r(y, \mathbf{x})$. We will use this to generate an output with the following procedure:

---
**Reward-Weighted Sampling**

---
**Require:** input $\mathbf{x}$, posterior predictive $p(y \mid \mathbf{x}, \mathcal{D})$, reward $r$, temperature $T$, number of samples $k$
1: **for** $i \leftarrow 1$ to $k$ **do**
2:     sample $y_i \sim p(y \mid \mathbf{x}, \mathcal{D})$
3:     $l_i \leftarrow \exp\big(r(y_i, \mathbf{x})/T\big)$
4: **end for**
5: $q_i \leftarrow l_i / \sum_{j=1}^{k} l_j \quad (i = 1, \dots, k)$
6: Draw $I \sim \text{Categorical}(q_1, \dots, q_k)$
7: **return** $y_{\text{out}} \leftarrow y_I$

---

For simplicity, we will assume a reward given by

$$r(y, \mathbf{x}) = -\left(y - \mathbf{w}_R \cdot \frac{\mathbf{x}}{\sqrt{d}}\right)^2. \qquad (6)$$

Note that $\mathbf{w}_R \neq \mathbf{w}_T$ in general. In realistic settings, this models the fact that LLM-as-a-Judge is not a perfect verifier. Hence, this settings will allow us to study the effect of the quality of the Judge on generalization error.

In this paper, we are interested in computing the generalization error of this model defined by

$$\delta = \mathbb{E}_{\mathbf{x}}(\delta(\mathbf{x})), \qquad (7)$$

$$\delta(\mathbf{x}) = \mathbb{E}_{y_1,\dots,y_k}\left[\frac{\sum_{i=1}^{k}\left(y_i - \mu_T(\mathbf{x})\right)^2 e^{-\left(y_i - \mu_R(\mathbf{x})\right)^2/T}}{\sum_{j=1}^{k} e^{-\left(y_j - \mu_R(\mathbf{x})\right)^2/T}}\right] \cdot \qquad (8)$$

Here we use the notation $\mu_T(\mathbf{x}) := \mathbf{w}_T \cdot \frac{\mathbf{x}}{\sqrt{d}}$, $\mu_R(\mathbf{x}) := \mathbf{w}_R \cdot \frac{\mathbf{x}}{\sqrt{d}}$.

## 3. Analysis Of The Generalization Error

In this section we analyze the high-dimensional behavior of the Bayesian regression model introduced above, with a particular focus on how inference-time sampling and the reward model shape the error.

### 3.1. Asymptotic Behavior Of The Generalization Error Via Deterministic Equivalents

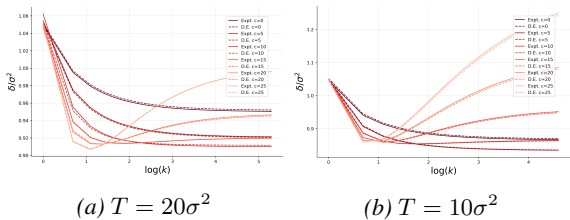

*(a)* $T = 20\sigma^2$        *(b)* $T = 10\sigma^2$

*Figure 2.* In the plot we have chosen $S = 1, \sigma = 10^{-4}, \gamma = 10^{-3}, n = 10^4, d = 10^1$ and sampled teacher weight $\mathbf{w}_T \sim \mathcal{N}(0, 2^2\mathbf{I})$. We have parameterized the reward weight as follows: $\mathbf{w}_R = (1 + cR/(R + S^2))\mathbf{w}_T$. Solid and dashed lines correspond to the experimental results and the formula in Result 1 respectively. On y-axis we plot generalization error normalized by its natural scale set by the noise $\delta/\sigma^2$ and in x-axis we plot the logarithm of the number of inference time samples $\log k$.

Before we get into the details of further theoretical analysis, we summarize the empirical findings in Figure 2 and compare with results of Result 1:

- When the reward is sufficiently accurate—formally, when $\|\mathbf{w}_R - \mathbf{w}_T\|/\|\mathbf{w}_T\|$ is small—the generalization error $\delta$ decreases *monotonically* with the number of inference-time samples $k$. We denote by $\mathcal{R}$ (highlighted in red in Figure 1) the region of reward weights exhibiting this monotonic behavior. Notably, within $\mathcal{R}$ the teacher reward is *not* optimal for fixed $(k, T)$; i.e., $\mathbf{w}_R = \mathbf{w}_T$ does not minimize $\delta$ (see Figure 2a).

- Comparing figures 2a–2b, we note that the set $\mathcal{R}$ shrinks as the temperature $T$ decreases (see also Figure 1). Outside this set, in its complement $\mathcal{R}^{\complement}$ (blue in Figure 1), Figure 2 shows that $\delta$ becomes *non-monotonic* in $k$, with a finite $k$ beyond which increasing $k$ worsens error. Consequently, for a fixed $\mathbf{w}_R$, lowering $T$ can induce a transition from $\mathcal{R}$ to $\mathcal{R}^{\complement}$; equivalently, at fixed $k$ there may exist an *optimal* temperature $T$ that minimizes $\delta$ (confirmed in Figure 5).

In the high dimensional setting, deterministic-equivalents simplify the predictive mean and variance by $m$ and $\Sigma = s^2$ as follows:

**Result 1.** *In the limit of $d, n \to \infty$, with $\alpha = d/n < 1$ fixed, for sufficiently small noise scale, i.e., there exists $\sigma_c(\alpha, R)$ (see Appendix B for an explicit expression) such that for $\sigma \ll \sigma_c(\alpha, R)$, the generalization error is given by*

$$\delta(\mathbf{x}) = \mathbb{E}\left[ \frac{\sum_{i=1}^{k} \left(y_i - \mu_T(\mathbf{x})\right)^2 e^{-\left(y_i - \mu_R(\mathbf{x})\right)^2 / T}}{\sum_{j=1}^{k} e^{-\left(y_j - \mu_R(\mathbf{x})\right)^2 / T}} \right],$$

(9)

*to the leading order in $\sigma$. Here the expectation is over $y_i \sim \mathcal{N}(m(\mathbf{x}), s(\mathbf{x})^2), i = 1, 2, \ldots, k$. The posterior predictive has a mean $m(\mathbf{x})$ and variance $\Sigma(\mathbf{x}) = s(\mathbf{x})^2$ as follows*

$$m(\mathbf{x}) = \frac{\mathbf{x}^\top}{\sqrt{d}} \mathbf{A}_R \mathbf{w}_T, \ s(\mathbf{x})^2 = \sigma^2 + \gamma^2 \frac{\mathbf{x}^\top}{\sqrt{d}} \mathbf{B}_R \frac{\mathbf{x}}{\sqrt{d}}.$$

(10)

*The matrices $\mathbf{A}_R, \mathbf{B}_R$ are defined in terms of renormalized ridge $R$ that satisfies*

$$R\left(1 - \alpha \, m_{\Sigma}(R)\right) = \hat{R} := \frac{\sigma^2}{\gamma^2} \alpha, \ m_{\Sigma}(R) := \frac{1}{d} Tr\left[\Sigma(\Sigma + RI)^{-1}\right].$$

(11)

*The solution of $R$ from above equation is used to define $\mathbf{A}_R, \mathbf{B}_R$ as follows*

$$\mathbf{A}_R := \Sigma(\Sigma + R\mathbf{I})^{-1}, \ \mathbf{B}_R := R(\Sigma + R\mathbf{I})^{-1} = \mathbf{I} - \mathbf{A}_R.$$

(12)

*Proof-sketch.* See appendix B for more details.

In this paper we will focus on the simple setup where $\Sigma = S^2 \mathbf{I}$. In this case we can explicitly solve for the renormalized ridge as follows

$$R = \frac{1}{2} S^2 \left( \frac{d}{n} + \frac{\hat{R}}{S^2} - 1 + \left( \left(1 - \frac{d}{n} - \frac{\hat{R}}{S^2}\right)^2 + \frac{4\hat{R}}{S^2} \right)^{\frac{1}{2}} \right).$$

(13)

In addition in this case the matrices $\mathbf{A}_R, \mathbf{B}_R$ are proportional to identity matrix

$$\mathbf{A}_R := \frac{S^2}{R + S^2} \mathbf{I}, \qquad \mathbf{B}_R := \frac{R}{R + S^2} \mathbf{I}.$$

(14)

These expressions are going to be useful in the later sections to have close form expression of generalization error $\delta$ in various parameter domains of interest.

## 3.2. High- And Low-temperature Behavior Of The Generalization Error

We now provide a theoretical account of these phenomena. In this section, we present two results that will help us gain insight later into generalization error behavior. Specifically, we analyze $\delta$ in two complementary regimes of the reward temperature: (i) a high-temperature (weak-reward)

expansion, where the selection reweighting is perturbative, and (ii) a low-temperature ("best-of-$k$") regime, where selection concentrates on high-reward samples and extreme-value effects dominate. The next two results formalize these regimes.

In the limit of an ample amount of data $d/n \to 0$ with a flat prior $\sigma/\gamma \to 0$, the temperature scale is controlled by $s^2 \approx \sigma^2$. In the high-temperature limit we present the following result.

**Result 2** (High-$T$ expansion). *For temperature much larger that the posterior predictive's variance, i.e., $T \gg s(\mathbf{x})^2$ the expectation value of the error can be organized as a series as follows*

$$\delta(\mathbf{x}) = \Delta_T(\mathbf{x})^2 + s^2(\mathbf{x})$$
$$+ \Sigma_{l=1}^{3} (-1)^l \frac{C_l(\mathbf{x})}{t(\mathbf{x})^l} \prod_{i=1}^{l} \left(1 - \frac{i}{k}\right) + \mathcal{O}\left(t(\mathbf{x})^{-4}\right)$$

(15)

*Where we have defined $t(\mathbf{x}) = \frac{T}{2s(\mathbf{x})^2}$ and*

$$C_l(\mathbf{x}) = 2 \Delta_T(\mathbf{x}) \Delta_R(\mathbf{x}) + s^2(\mathbf{x}) + (l-1) \Delta_R(\mathbf{x})^2$$
$$\Delta_T(\mathbf{x}) := m(\mathbf{x}) - \mu_T(\mathbf{x}), \quad \Delta_R(\mathbf{x}) := m(\mathbf{x}) - \mu_R(\mathbf{x}).$$

(16)

*All other quantities are as in Result 1.*

*Proof-sketch.* Let $z$ denote the partition function over $k$ i.i.d. draws from $p(y|\mathbf{x}, \mathcal{D})$ with quadratic reward; expand $\mathbb{E} \log z$ around $\mathbb{E}z$ via a controlled the cumulant expansion for $t \gg 1$. The $1/t, 1/t^2$ and $1/t^3$ terms produce the $C_1(\mathbf{x}), C_2(\mathbf{x})$ and $C_3(\mathbf{x})$ structure; substituting the deterministic equivalents for $m, \Sigma$ converts it to the form mentioned in the Result. See Appendix C for the details.

In Appendix F, we show that the structure of the perturbation theory remains the same for a generic probabilistic model with a general reward. We also discuss statistical significance of various terms in the expansion.

Now we turn to best-of-$k$ setting, given by the $T \to 0$ limit, and present our theoretical finding below.

**Result 3** (Low-$T$ best-of-$k$ sampling). *When we have access to the exact teacher weight $\mathbf{w}_R = \mathbf{w}_T = \mathbf{w}$, the leading order result for $T \to 0$ followed by $k \to \infty$ is given by*

$$\delta(\mathbf{x}) = \frac{\pi}{k^2} \left[ s^2(\mathbf{x}) \, \exp\left( \frac{\Delta_T(\mathbf{x})^2}{s^2(\mathbf{x})} \right) \right]$$

(17)

*All the quantities are as in Result 1 and Result 2.*

*Proof-sketch.* At $T = 0$, the softmax reduces to a minimum of chi-squared random variables. This is governed by the Weibull distribution at large $k$ according to extreme value theory. Finally, substituting the deterministic equivalents for $m, \Sigma$ and evaluating the expectation value gives the generalization error mentioned in the Result. See Appendix 11 for details.

Note that the low-temperature scaling of $\delta$ with the number of inference time samples $k$ is independent of the amount of the amount of training data. These theoretical results are compared with the experiment in Figure 6.

In Appendix F, we establish that in general depending on the coverage coefficient $\beta$ of the predictive distribution near the target one can get more general scaling law $\Theta(k^{-\frac{2}{1+\beta}})$. For the Gaussian predictive distribution one gets the $\Theta(k^{-2})$ law as in the Result 3.

### 3.3. Optimal Reward May Differ From The Teacher

Naively, one might expect that the optimal reward, one that leads to the best generalization error, is given by the teacher itself. However, Figure 3 shows that this may not always be true. Here, we compare the generalization error achieved when the reward weight equals the teacher ($\mathbf{w}_R = \mathbf{w}_T$) versus when it differs, across different values of $k$ and $T$. The plot reveals a consistent pattern: when $\mathbf{w}_R$ is close to $\mathbf{w}_T$, the error is *lower* when the reward weight is *slightly shifted* away from the teacher. This additional shift required for the optimal reward grows systematically with the temperature scale $T$.

We can get insight into this behavior exploiting our Result 2. By setting the first derivative of $\delta(\mathbf{x})$, given in Result 2, with respect to $\mathbf{w}_R$ to zero, we arrive at the following conclusion:

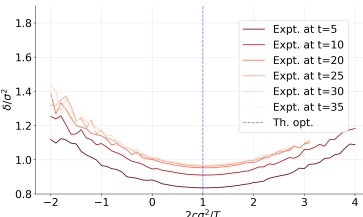

*Figure 3.* In the plot we have chosen $S = 1, \sigma = 10^{-4}, \gamma = 10^{-3}, n = 10^4, d = 10, k = 50$ and used the following parameterization $\mathbf{w}_R = (1 + cR/(R + S^2))\mathbf{w}_T$ and sampled $\mathbf{w}_T \sim \mathcal{N}(0, 2^2\mathbf{I})$. This plot shows dependence of $\delta$ on $c$ for various values of $T$ at fixed $k$. We see that $\delta$ is minimized at $c \approx T/(2\sigma^2)$ as expected from Remark 2.

*Remark* 2. There exists an optimal reward weight that differs from the teacher weight by the following formula

$$\mathbf{w}_R(\mathbf{x}) = \mathbf{w}_T + \left(\frac{k}{k-2}t(\mathbf{x})\right)\mathbf{B}_R\,\mathbf{w}_T \qquad (18)$$

This formula provides a controlled approximation in the domain stated in Result 2 as long as

$$\frac{\|\mathbf{w}_R(\mathbf{x}) - \mathbf{w}_T\|}{\|\mathbf{w}_T\|} \ll 1. \qquad (19)$$

The Remark also quantifies the empirically observed fact

that as $T$ increases, the optimal $\mathbf{w}_R$ moves proportionally away from $\mathbf{w}_T$.

### 3.4. There Exists An Optimal Number For Inference-time Samples

Figure 4 shows that when the reward weight $\mathbf{w}_R$ is sufficiently misaligned from the teacher $\mathbf{w}_T$, the test error as a function of the number of inference samples $k$ is *non-monotonic*: it first decreases (benefiting from a better draw among $k$ candidates) and then *increases* beyond an optimal value of $k$. We can get some more insight into this behavior by the following Remark.

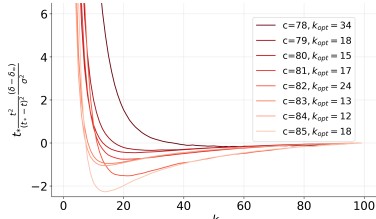

*Figure 4.* In the plot we have chosen $S = 1, \sigma = 10^{-4}, \gamma = 10^{-3}, n = 10^4, d = 10, T = 200\sigma^2$ and used the following parameterization $\mathbf{w}_R = (1 + cR/(R + S^2))\mathbf{w}_T$ and sampled $\mathbf{w}_T \sim \mathcal{N}(0, 2^2\mathbf{I})$. We plot the scaled value of $\delta - \delta_\infty, \delta_\infty \approx \delta_{k=100}$ as a function of $k$ for various values of $c$. This shows existence of an optimal value of $k$ - theoretical prediction for it is denoted as $k_{opt}$ as given in Remark 3.

*Remark* 3. There exists an optimal value of inference samples $k$, when Result 2 is valid and

$$t < \frac{3C_2(\mathbf{x})}{C_1(\mathbf{x})} \equiv t_*, \quad C_1(\mathbf{x}) > 0, \quad C_2(\mathbf{x}) > 0 \qquad (20)$$

In this case, as we increase $k$ the error decreases until when we reach

$$k = \left\lceil \frac{4}{3}\frac{t_*}{t_* - t} \right\rceil \approx \frac{2t_*}{t_* - t} \qquad (21)$$

and far beyond this, increase in $k$ increases the error. For larger values of $t$, increase in $k$ always decreases error.

*Proof-sketch.* This is obtained by setting the first derivative of $\delta(\mathbf{x})$, given in Result 2, with respect to $k$ to zero.

Note that above Remark only applies when $t \gg 1$ and $t < 3C_2/C_1$. But when $\mathbf{w}_R$ is sufficiently close to $\mathbf{w}_T$, $C_2/C_1 \sim 1$. When $\mathbf{w}_R$ is sufficiently close to $\mathbf{w}_T$, increase in $k$ decreases the error $\delta$ since in this case above Remark does not apply. Whereas when $\mathbf{w}_R$ is sufficiently far from $\mathbf{w}_T$ ($C_2/C_1$ becomes larger since $\Delta_R$ grows), above Remark shows existence of an optimal value for $k$.

### 3.5. There Exists An Optimal Temperature For Reward Sampling

Figure 5 examines generalization $\delta$ error as a function of temperature $T$ at fixed number of inference time samples $k$.

Empirically, the generalization error exhibits a clear local minimum around a critical value of $T$, rather than decreasing or increasing monotonically. Interpreting this through the high-$T$ expansion, the minimum corresponds to a particular balance between the first- and second-order correction terms governed by $C_1$ and $C_2$. Temperature $T$ controls how sharply the selection favors high-reward samples among the $k$ candidates. At very high temperatures, selection is nearly uniform and the benefits of the reward model are muted; at very low temperatures, selection becomes too aggressive and can over-amplify any mismatch between $\mathbf{w}_R$ and $\mathbf{w}_T$, increasing error. The optimal temperature thus trades off these effects. It scales linearly with $\Sigma$ and grows with misspecification via $C_2/C_1$. The location of the optimal temperature for given $k, \mathbf{w}_R$ is determined from the theoretical result below:

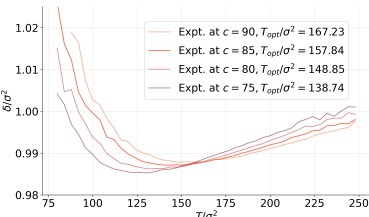

*Figure 5.* In the plot we have chosen $S = 1, \sigma = 10^{-4}, \gamma = 10^{-3}, n = 10^4, d = 10$ and used the following parameterization $\mathbf{w}_R = (1 + cR/(R + S^2))\mathbf{w}_T$ and sampled $\mathbf{w}_T \sim \mathcal{N}(0, 2^2\mathbf{I})$. This plot shows existence of an optimal value of temperature $T$ at fixed $k = 50$. For the optimal value, we find reasonable agreement with the theoretical prediction in in Remark 4.

*Remark* 4. For a given number of inference-time samples $k > 2$, training dataset size $n$ and reward weight $\mathbf{w}_R$, there exists an optimal temperature for the rewarding process

$$t(\mathbf{x}) = 2\left(1 - \frac{2}{k}\right)\frac{C_2(\mathbf{x})}{C_1(\mathbf{x})}, \quad C_1(\mathbf{x}) > 0, \quad C_2(\mathbf{x}) > 0 \tag{22}$$

This formula is valid in the domain stated in Result 2.

*Proof-sketch.* This is obtained by setting the first derivative of $\delta(\mathbf{x})$, given in Result 2, with respect to $t$ to zero.

### 3.6. Teacher-rewarded Best-of-$k$ Shows Power-law Decay

The low-$T$ result in Result 3 shows an inverse–quadratic $k^{-2}$ decay of the error when the reward matches the teacher ($\mathbf{w}_R = \mathbf{w}_T$). Here we sharpen that statement by identifying a concrete, practically relevant parameter domain in which the leading-order constant in front of $k^{-2}$ can be written in closed form. Explicit formula clarifies how dimensionality, sample size, noise, and prior scale combine in the low-temperature limit.

*Remark* 5. As a refinement of Result 3, consider the parameter regime

$$\frac{\gamma^2}{d}\mathrm{Tr}(\mathbf{B}_R\Sigma) \ll \sigma^2.$$

In the low-temperature limit $T \to 0$ followed by $k \to \infty$, the leading-order generalization error for $\mathbf{w}_R = \mathbf{w}_T = \mathbf{w}$ is given by

$$\delta = \frac{\pi\sigma^2}{k^2}\frac{1}{\sqrt{1 - \frac{2}{\sigma^2 d}\mathbf{u}^\top\Sigma\mathbf{u}}}, \quad \mathbf{u} := \mathbf{B}_R\mathbf{w}$$

*Proof-sketch.* In this domain the $\mathbf{x}$-dependence of $s^2(\mathbf{x})$ is small relative to $\sigma^2$ and this allows us to reliably set $s^2(\mathbf{x}) \approx \sigma^2$ in Result 3 to evaluate the expectation value.

In the flat prior limit, i.e, $\gamma^2 \gg \sigma^2$, this regime corresponds to ample amount of data per dimension, i.e., $n \gg d$. For the isotropic sample covariance $\Sigma = S^2\mathbf{I}$ that we are analyzing in this paper, we have $\mathbf{u} = \frac{R}{R+S^2}\mathbf{w}$. In the limit of flat prior with ample amount of data, this further simplifies to $\mathbf{u} \approx (1/S^2)(\sigma^2/\gamma^2)(d/n)\mathbf{w}$. The Remark above shows as task difficulty increases, i.e, $\sigma$ gets larger keeping other parameters fixed, generalization error $\delta$ and even the scaled generalization error $\delta/\sigma^2$ increases.

### 3.7. Trade-off Between Training And Inference-time Compute

In practice we often face a budget allocation decision: should additional compute be spent on *training* (e.g., acquiring/processing more samples $n$) or on *inference-time* (e.g., drawing more candidates $k$ and selecting via the reward)? When the reward is well aligned with the teacher and we operate in the low-temperature regime, best-of-$k$ style selection can substantially reduce error with relatively modest inference cost. The question is how this compares with the addition of more data.

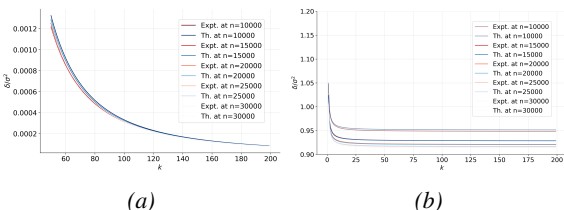

$(a)$          $(b)$

*Figure 6.* In the plot we have chosen $S = 1, \sigma = 10^{-4}, \gamma = 10^{-3}, d = 10$. (a) We plot the theoretical (given in Result 3) and experimental value of $\delta$ at $T = 0$ and find good agreement for $\mathbf{w}_R = \mathbf{w}_T \sim \mathcal{N}(0, 2^2\mathbf{I})$. We see that in this domain scaling $n$ higher is less useful compared to scaling $k$. (b) We plot the theoretical (given in Result 2) and experimental value of $\delta$ and find good agreement for $\mathbf{w}_R = \mathbf{w}_T \sim \mathcal{N}(0, 2^2\mathbf{I})$. We see that in this domain scaling $n$ higher is more useful compared to scaling $k$.

*Remark* 6. Given access to exact teacher weight, it is beneficial to scale inference-time compute over adding more

training samples in the following regime: consider $T \to 0$ followed by $k \to \infty$ with

$$\frac{\gamma^2}{d}\text{Tr}(\mathbf{B}_R\mathbf{\Sigma}) \ll \sigma^2, \quad R \ll \sigma^2 \tag{23}$$

That is within this domain,

$$\frac{\partial \log \delta}{\partial \log k} = -2, \qquad \frac{\partial \log \delta}{\partial \log n} = -\frac{\alpha\partial_\alpha\left(\mathbf{u}^\top\mathbf{\Sigma}\mathbf{u}\right)}{\sigma^2 d - 2\mathbf{u}^\top\mathbf{\Sigma}\mathbf{u}} \tag{24}$$

and hence

$$\left|\frac{\partial \log \delta}{\partial \log k}\right| \gg \left|\frac{\partial \log \delta}{\partial \log n}\right| \tag{25}$$

*Proof-sketch.* See Appendix E for details.

In the flat prior, i.e, $\gamma^2 \gg \sigma^2$, ample data limit, i.e., $n \gg d$, the second condition in the Remark quantifies prior quality - roughly speaking it dictates that when the prior $\gamma$ is broad enough, inference time compute is beneficial over training time compute. For the isotropic sample covariance $\mathbf{\Sigma} = S^2\mathbf{I}$, putting back explicit formula for $\mathbf{u} \approx R/S^2\mathbf{w} \approx (1/S^2)(\sigma^2/\gamma^2)(d/n)\mathbf{w}$ shows that

$$\frac{\partial \log \delta}{\partial \log n} = -2\frac{\frac{\mathbf{w}\cdot\mathbf{w}}{d}\frac{1}{S^2}\frac{d^2\sigma^2}{n^2\gamma^4}}{1 - 2\frac{\mathbf{w}\cdot\mathbf{w}}{d}\frac{1}{S^2}\frac{d^2\sigma^2}{n^2\gamma^4}} \tag{26}$$

In this case, under an even weaker condition $R^2 < \sigma^2$ already we see that scaling inference time is beneficial over scaling training compute. If $\sigma$ increases keeping other parameters held fixed, the magnitude of the derivative of $\log \delta$ w.r.t. $\log n$ increases. Hence as the task becomes more difficult the advantage of inference time scaling degrades. The same statement holds true if $\gamma$ decreases while other parameters are held fixed.

We empirically validate these results on advantages of inference-time scaling over training compute in Figure 6a. It is clear from Figure 6a that fractional increase in $k$ decreases generalization error $\delta$ more compared to the same fractional increase in $n$ within the domain of parameters considered in the plot. However, inference-time scaling is not always advantageous over increasing training compute - we explain this in Figure 6b. We conclude that when we have access to a good quality reward model and the task is easy enough, the addition of inference time compute is beneficial over additional training compute.

## 4. Qualitative Agreement With Large Language Model Reasoning

In this section, we discuss implications of our theoretical results for inference time scaling of large language models such as Llama-3-8B-Instruct, DeepSeek-R1-Distill-Qwen-1.5B on openai/gsm8k validation dataset (prompt included 8 chain of thought demonstrations). As the Judge model we

used Mistral-7B-Instruct-v0.3 or Qwen2.5-Math-PRM-7B. For the details of experimental design see Appendix H.2.

In the linear model we have observed that the when the reward is not close to the teacher model there exists an optimal value of inference time samples. This fact has been observed in large language models in (Snell et al., 2025; Chen et al., 2024a). Figure 7 shows that there is a global minima in generalization error as a function of the inference time samples at fixed temperature, qualitatively validating our theoretical observation.

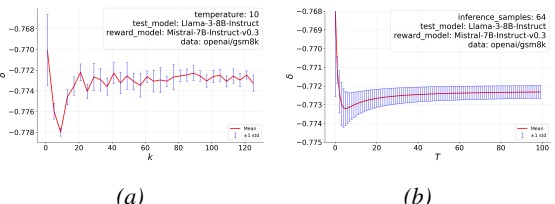

*(a)*            *(b)*

*Figure 7.* In the plot we have generated inference time samples from Llama-3-8B-Instruct. For each question and response pair we used Mistral-7B-Instruct-v0.3 (with simple/weak prompt). For generalization error we used the definition in equation 27. The variance in the graph is due to non-zero temperature of generation in Llama-3-8B-Instruct. Plot (a), (b) shows existence of an optimal value of $k, T$ respectively.

Our second observation is that there exists an optimal temperature for reward sampling. To the best of our knowledge this is a new observation and in this section we present experiments supporting it. For a given question $\mathbf{x}$ we generate $k$ responses $y_i, i = 1, 2, \ldots, k$ from the large language model under study and use a judge language model to assign a reward $r(y_i, \mathbf{x})$ to $i$ th response. The generalization error is defined by

$$\delta = -\mathbb{E}_{\mathbf{x}} \sum_{i=1}^{k} \frac{e^{\frac{r(y_i,\mathbf{x})}{T}}}{\sum_{j=1}^{k} e^{\frac{r(y_j,\mathbf{x})}{T}}} \, v(y_i, \mathbf{x}) \tag{27}$$

Here $v(y_i, \mathbf{x}) \in \{0, 1\}$ is 1 when the response is correct and 0 otherwise. When $T = 0$ this reduces to the best of $k$ rewarding process. In Figure 7 we present experimental results confirming that there exists an optimal value of $T$.

In Figure 8 we study the change of optimal values of $k, T$ when the judge is changed from a weaker one to a stronger one by keeping the judge-LLM the same, but using a detailed prompt. During our study of the toy model, from Figure 4 we see that for a range of $c$, optimal value of $k$ remained almost fixed. However, as the reward model became stronger, i.e., $c$ is decreased, Figure 5 showed that optimal $T$ also decreased. We see similar qualitative behavior from Figure 8a and 8b respectively.

In Figure 9 we use Qwen2.5-Math-PRM-7B as the reward model. This verifies each step of the reasoning and corresponds to nearly ideal reward. The plot shows that for

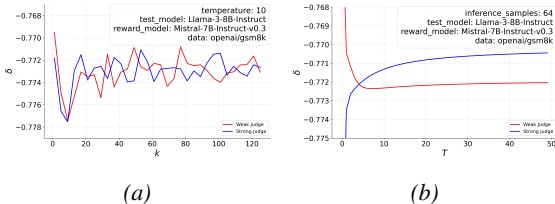

*(a)*          *(b)*

*Figure 8.* We use the same setup as in Figure. 7 with two different prompts for the reward model - a simple/weak prompt and a detailed/strong prompt (see the associated code for the details of the prompt). We call these weak and strong judge respectively. Plot (a), (b) shows behavior of the optimal value of $k, T$ respectively under the change of judge. In the plot we have used a single generation from the Llama-3-8B-Instruct and kept the judge deterministic.

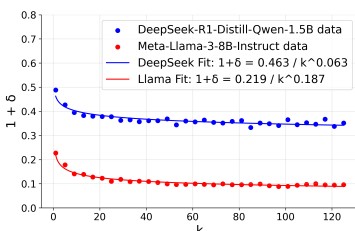

*Figure 9.* In this plot we plot the error $1 + \delta$ of various base models vs $k$ at fixed reward temperature $T = 0$ for nearly ideal reward model Qwen2.5-Math-PRM-7B. It is clear from the plot that the curves are well fitted by an effective scaling law similar to the result in Remark 5 except that the exponent of $k$ is different that can be explained in the more general framework of Appendix F.

best-of-$k$ method the error decreases as a power law in inference time samples (almost monotonically). Similar result is established theoretically in Remark 5. Note that the judge in our case takes into account the reasoning process, not just final result of being correct or wrong and our metric is much different compared to pass@k used in (Brown et al., 2024; Schaeffer et al., 2025a).

## 5. Conclusion and Limitations

We introduce a simple and analytically tractable model for studying inference-time scaling, in which multiple candidate outputs are generated and selected through a quadratic reward-based mechanism. Our analysis shows that when the reward model is sufficiently well aligned with the task, increasing the number of inference-time samples leads to a reliable improvement in generalization. By contrast, when the reward model is substantially misspecified, the generalization error becomes non-monotonic, yielding both an optimal finite number of samples and an optimal selection temperature. In the best-of-k regime with an ideal reward model, we further identify a power-law scaling of the generalization error. We complement the theory with experiments on large language models evaluated on a mathematical reasoning benchmark, which provide qualitative support for

the predicted phenomena. To address the simplified assumptions adopted in the main text, Appendix F discusses how the framework extends more generally. At the same time, the LLM experiments should be interpreted as qualitative validations rather than literal recovery of theoretical parameters. Although Appendix F outlines the formalism for heavy-tailed and potentially multimodal predictive distributions, a careful analysis of these settings remains an open problem.

We also show that the reward model achieving optimal performance need not coincide with the teacher model. By exploiting structural properties of the trained predictor, we derive the corresponding optimal reward model within our framework. This suggests a possible analogue for large language models: an auxiliary classifier could be used to identify systematic weaknesses of the base model and to determine which queries should be evaluated by the reward model, thereby enabling a form of classifier-guided reward shaping. A thorough investigation of this direction is left for future work.

In addition, we study the trade-off between training-time and inference-time scaling. Extending this analysis in a careful and realistic manner to large language models remains an important open question.

Finally, the present work is limited to the i.i.d. sampling setting. In Appendix G, we briefly outline how the framework may be extended to non-i.i.d. sampling procedures, such as beam search and related decoding strategies.

## Acknowledgments

I.H. is supported by DARPA grant AIQ-HR00112520041. C.P. is supported by an NSF CAREER Award (IIS-2239780), DARPA grants DIAL-FP-038 and AIQ-HR00112520041, the Simons Collaboration on the Physics of Learning and Neural Computation, and the William F. Milton Fund from Harvard University. I.H. and C.P. thank the Stanford Institute for Human-Centered Artificial Intelligence and the Kavli Institute for Theoretical Physics for its hospitality during the completion of this work. This work has been made possible in part by a gift from the Chan Zuckerberg Initiative Foundation to establish the Kempner Institute for the Study of Natural and Artificial Intelligence.

## Impact Statement

This paper presents work whose goal is to advance the field of machine learning in particular inference time optimization of large language models.

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

# A. Review of extreme value statistics

## A.1. Limit Laws for Maxima

Here we note some of the useful results in extreme value theory from (de Haan and Ferreira, 2007).

**Result 4** (Fisher–Tippett–Gnedenko). *Let $X_1, X_2, \ldots$ be i.i.d. non-degenerate random variables with distribution function $F$, i.e, $F(x) = \mathbb{P}(X \leq x)$, and let $M_n = \max\{X_1, \ldots, X_n\}$. If there exist normalising constants $a_n > 0$ and $b_n \in \mathbb{R}$ and a non-degenerate distribution function $H$ such that*

$$\mathbb{P}\left(\frac{M_n - b_n}{a_n} \leq x\right) \xrightarrow[n\to\infty]{} H(x)$$

*then $H$ must be (up to affine change of variable) one of the three extreme value distributions:*

$$\textit{Fréchet } (\alpha > 0): \quad \Phi_\alpha(x) = \begin{cases} 0, & x \leq 0, \\ \exp\{-x^{-\alpha}\}, & x > 0, \end{cases}$$

$$\textit{Weibull } (\alpha > 0): \quad \Psi_\alpha(x) = \begin{cases} \exp\{-(-x)^\alpha\}, & x \leq 0, \\ 1, & x > 0, \end{cases}$$

$$\textit{Gumbel}: \quad \Lambda(x) = \exp\{-e^{-x}\}, \qquad x \in \mathbb{R}.$$

## A.2. Maximum Domains of Attraction and Norming Constants

**Definition 5** (Maximum domain of attraction). We say $F$ belongs to the maximum domain of attraction of $H$ (write $F \in \mathrm{MDA}(H)$) if there exist constants $a_n > 0, b_n \in \mathbb{R}$ such that

$$\lim_{n\to\infty} F^n(a_n x + b_n) = H(x)$$

*Remark* 7 (Characterisation via exceedance rates). Let $H$ be a (standard) extreme value distribution. Then $F \in$

$\mathrm{MDA}(H)$ with norming constants $a_n > 0, b_n \in \mathbb{R}$ if and only if

$$\lim_{n\to\infty} n\big(1 - F(a_n x + b_n)\big) = -\ln H(x)$$

For later convenience we define right endpoint $x_F := \sup\{x : F(x) < 1\}$, complementary distribution function $\bar{F}(x) := 1 - F(x)$ and quantile function $F^\leftarrow(t) = \inf\{x \in \mathbb{R} : F(x) \geq t\}$.

### A.2.1. THE MAXIMUM DOMAIN OF ATTRACTION OF THE FRÉCHET DISTRIBUTION

**Result 6** (MDA of Fréchet). *Let $F$ have a finite right endpoint $x_F = \infty$ and and assume there exists $z < x_F$ such that $F$ is differentiable in $(z, x_F)$ The following statements are equivalent:*

(i) *$F$ satisfies* von Mises condition, *i.e.,*

$$\lim_{x\to x_F-} \frac{x\, F'(x)}{1 - F(x)} = \alpha > 0 \qquad (28)$$

(ii) *$F \in \mathrm{MDA}(\Phi_\alpha)$ with a possible choice of norming constants*

$$b_n = 0, \qquad a_n = F^\leftarrow\left(1 - \frac{1}{n}\right),$$

### A.2.2. THE MAXIMUM DOMAIN OF ATTRACTION OF THE WEIBULL DISTRIBUTION

**Result 7** (MDA of Weibull). *Let $F$ have a finite right endpoint $x_F < \infty$ and and assume there exists $z < x_F$ such that $F$ is differentiable in $(z, x_F)$ The following statements are equivalent:*

(i) *$F$ satisfies* von Mises condition, *i.e.,*

$$\lim_{x\to x_F-} \frac{(x_F - x)\, F'(x)}{1 - F(x)} = \alpha > 0 \qquad (29)$$

(ii) *$F \in \mathrm{MDA}(\Psi_\alpha)$ with a possible choice of norming constants*

$$b_n = x_F, \qquad a_n = x_F - F^\leftarrow\left(1 - \frac{1}{n}\right),$$

### A.2.3. THE MAXIMUM DOMAIN OF ATTRACTION OF THE GUMBEL DISTRIBUTION

**Result 8** (MDA of Gumbel). *Let $F$ be a distribution with right endpoint $x_F \leq \infty$, and assume there exists $z < x_F$ such that $F$ is at least twice differentiable in $(z, x_F)$. Define auxiliary function $a(x) = \bar{F}(x)/F'(x)$. The following statements are equivalent:*

(i) $F$ is a von Mises function, i.e.,

$$\lim_{x \to x_F-} a'(x) = 0$$

(ii) $F \in MDA(\Lambda)$ with a possible choice of norming constants

$$b_n = F^{\leftarrow}(1 - 1/n), \qquad a_n = a(b_n)$$

## B. Proof of Result 1

**Result 9.** *In the limit of $d, n \to \infty$, with $\alpha = d/n < 1$ fixed, for sufficiently small noise scale, i.e., there exists $\sigma_c(\alpha, R)$ such that for $\sigma \ll \sigma_c(\alpha, R)$, the generalization error is given by*

$$\delta = \mathbb{E}_{\mathbf{x}} \mathbb{E}_{y_1, y_2, \dots} \left[ \frac{\sum_{i=1}^{k} (y_i - \mu_T(\mathbf{x}))^2 e^{-(y_i - \mu_R(\mathbf{x}))^2/T}}{\sum_{j=1}^{k} e^{-(y_j - \mu_R(\mathbf{x}))^2/T}} \right], \tag{30}$$

*to the leading order in $\sigma$. Here the expectation is over $\mathbf{x} \sim \mathcal{N}(0, \boldsymbol{\Sigma})$, $y_i \sim \mathcal{N}(m(\mathbf{x}), s(\mathbf{x})^2)$, $i = 1, 2, \dots, k$. The posterior predictive has mean $m(\mathbf{x})$ and variance $\Sigma(\mathbf{x}) = s(\mathbf{x})^2$ as follows*

$$m(\mathbf{x}) = \frac{\mathbf{x}}{\sqrt{d}}^\top \mathbf{A}_R \mathbf{w}_T, \quad s(\mathbf{x})^2 = \sigma^2 + \gamma^2 \frac{\mathbf{x}}{\sqrt{d}}^\top \mathbf{B}_R \frac{\mathbf{x}}{\sqrt{d}}. \tag{31}$$

*The matrices $\mathbf{A}_R, \mathbf{B}_R$ are given by*

$$\mathbf{A}_R := \boldsymbol{\Sigma}(\boldsymbol{\Sigma} + R\mathbf{I})^{-1}, \qquad \mathbf{B}_R := R(\boldsymbol{\Sigma} + R\mathbf{I})^{-1} = \mathbf{I} - \mathbf{A}_R. \tag{32}$$

*and the renormalized ridge $R$ is given by*

$$\hat{R} = R\left(1 - \alpha \, m_{\boldsymbol{\Sigma}}(R)\right) = \frac{\sigma^2}{\gamma^2} \alpha,$$
$$m_{\boldsymbol{\Sigma}}(R) := \frac{1}{d} Tr\left[\boldsymbol{\Sigma}(\boldsymbol{\Sigma} + R I)^{-1}\right]. \tag{33}$$

*Proof-sketch.* Let the empirical covariance be $\widehat{\boldsymbol{\Sigma}} := \frac{1}{n} \sum_{i=1}^{n} \mathbf{x}^i \mathbf{x}^{i\top}$, then

$$\boldsymbol{\Omega} = \sigma^2 \alpha \left(\widehat{\boldsymbol{\Sigma}} + \hat{R}\mathbf{I}\right)^{-1}, \qquad \hat{R} := \frac{\sigma^2}{\gamma^2} \alpha. \tag{34}$$

The mean of the predictive is given by

$$m(\mathbf{x}) = \boldsymbol{\mu}^\top \frac{\mathbf{x}}{\sqrt{d}} = \frac{1}{\sigma^2} \frac{\mathbf{x}}{\sqrt{d}}^\top \boldsymbol{\Omega} \sum_{i=1}^{n} y^i \frac{\mathbf{x}^i}{\sqrt{d}} \tag{35}$$

$$= \frac{\mathbf{x}}{\sqrt{d}}^\top \left(\mathbf{I} - \frac{1}{\gamma^2}\boldsymbol{\Omega}\right) \mathbf{w}_T + \frac{1}{\sigma^2} \frac{\mathbf{x}}{\sqrt{d}}^\top \boldsymbol{\Omega} \sum_{i=1}^{n} \eta^i \frac{\mathbf{x}^i}{\sqrt{d}}, \tag{36}$$

Now we use $\mathbf{I} - (1/\gamma^2)\boldsymbol{\Omega} = \mathbf{I} - \hat{R}\left(\widehat{\boldsymbol{\Sigma}} + \hat{R}\mathbf{I}\right)^{-1} = \widehat{\boldsymbol{\Sigma}}\left(\widehat{\boldsymbol{\Sigma}} + \hat{R}\mathbf{I}\right)^{-1}$ to get

$$m(\mathbf{x}) = \left\langle \widehat{\boldsymbol{\Sigma}}(\widehat{\boldsymbol{\Sigma}} + \hat{R}\mathbf{I})^{-1} \mathbf{w}_T, \frac{\mathbf{x}}{\sqrt{d}} \right\rangle + \frac{1}{\sigma^2} \frac{\mathbf{x}}{\sqrt{d}}^\top \boldsymbol{\Omega} \sum_{i=1}^{n} \eta^i \frac{\mathbf{x}^i}{\sqrt{d}}, \tag{37}$$

Conditioned test data $[\mathbf{x}^1, \dots, \mathbf{x}^n]$ and the test $\mathbf{x}$, the vector $\sum_{i=1}^{n} \eta^i \frac{\mathbf{x}^i}{\sqrt{d}}$ has zero mean with conditional covariance $\sigma^2 \sum_{i=1}^{n} \frac{\mathbf{x}^i}{\sqrt{d}} \frac{\mathbf{x}^i}{\sqrt{d}}^\top$. When $\sigma$ is sufficiently small we can ignore this contribution. We will present a detailed justification of this fact below.

The variance of the predictive can be simplified to

$$s^2(\mathbf{x}) = \frac{\mathbf{x}}{\sqrt{d}}^\top \boldsymbol{\Omega} \frac{\mathbf{x}}{\sqrt{d}} + \sigma^2 = \frac{\mathbf{x}}{\sqrt{d}}^\top \left[\sigma^2 \alpha (\widehat{\boldsymbol{\Sigma}} + \hat{R}\mathbf{I})^{-1}\right] \frac{\mathbf{x}}{\sqrt{d}} + \sigma^2$$

$$= \gamma^2 \frac{\mathbf{x}}{\sqrt{d}}^\top \left[\hat{R}(\widehat{\boldsymbol{\Sigma}} + \hat{R}\mathbf{I})^{-1}\right] \frac{\mathbf{x}}{\sqrt{d}} + \sigma^2, \tag{38}$$

There exists a $R > 0$ as defined in the Result such that, for any vectors $u, v$ with bounded norms,

$$u^\top \left[\widehat{\boldsymbol{\Sigma}}(\widehat{\boldsymbol{\Sigma}} + \hat{R}\mathbf{I})^{-1}\right] v \xrightarrow{p} u^\top \left[\boldsymbol{\Sigma}(\boldsymbol{\Sigma} + R\mathbf{I})^{-1}\right] v = u^\top \mathbf{A}_R v, \tag{39}$$

$$u^\top \left[\hat{R}(\widehat{\boldsymbol{\Sigma}} + \hat{R}\mathbf{I})^{-1}\right] v \xrightarrow{p} u^\top \left[R(\boldsymbol{\Sigma} + R\mathbf{I})^{-1}\right] v = u^\top \mathbf{B}_R v. \tag{40}$$

Applying equation 39 to equation 37, yields

$$m(\mathbf{x}) \xrightarrow{p} \frac{\mathbf{x}}{\sqrt{d}}^\top \mathbf{A}_R \mathbf{w}_T$$

Applying equation 40 to equation 38 with $u = v = \frac{\mathbf{x}}{\sqrt{d}}$ yields

$$s^2(\mathbf{x}) \xrightarrow{p} \sigma^2 + \gamma^2 \frac{\mathbf{x}}{\sqrt{d}}^\top \mathbf{B}_R \frac{\mathbf{x}}{\sqrt{d}}.$$

Now we turn to give a precise condition for when it is suitable to drop the noise term. Define

$$Z(\mathbf{x}) := \frac{1}{\sigma^2} \frac{\mathbf{x}}{\sqrt{d}}^\top \boldsymbol{\Omega} \sum_{i=1}^{n} \eta^i \frac{\mathbf{x}^i}{\sqrt{d}}. \tag{41}$$

Then, in the proportional limit $d, n \to \infty$ with $\alpha$ fixed, the variance of the label–noise term $Z(\mathbf{x})$ admits the deterministic equivalent

$$\text{Var}(Z(\mathbf{x})) \xrightarrow{d,n \to \infty} \sigma^2 \frac{\alpha \, m_{\boldsymbol{\Sigma}}^{(2)}(R)}{1 - \alpha \, m_{\boldsymbol{\Sigma}}^{(2)}(R)}, \tag{42}$$

$$m_{\boldsymbol{\Sigma}}^{(2)}(R) := \frac{1}{d} Tr\left[\boldsymbol{\Sigma}^2 (\boldsymbol{\Sigma} + R\mathbf{I})^{-2}\right]$$

Next we explain this. Conditioned on the training inputs $\{\mathbf{x}^i\}_{i=1}^n$ and the test point $\mathbf{x}$. Using equation 41 and the fact that $\boldsymbol{\eta} = (\eta^1, \ldots, \eta^n)^\top \sim \mathcal{N}(0, \sigma^2 \mathbf{I}_n)$ is independent of $\mathcal{D}$ and $\mathbf{x}$, we have

$$Z(\mathbf{x}) = \frac{1}{\sigma^2} \frac{\mathbf{x}}{\sqrt{d}}^\top \boldsymbol{\Omega} \sum_{i=1}^n \eta^i \frac{\mathbf{x}^i}{\sqrt{d}} = \frac{1}{\sigma^2} \frac{\mathbf{x}}{\sqrt{d}}^\top \boldsymbol{\Omega} \widetilde{X}^\top \boldsymbol{\eta},$$

where $\widetilde{X} \in \mathbb{R}^{n \times d}$ has rows $\frac{\mathbf{x}^i}{\sqrt{d}}^\top$. Hence

$$\mathbb{E}\big[Z(\mathbf{x}) \mid \mathcal{D}, \mathbf{x}\big] = 0,$$

and

$$\mathrm{Cov}\big(\widetilde{X}^\top \boldsymbol{\eta} \mid \mathcal{D}\big) = \mathbb{E}\big[\widetilde{X}^\top \boldsymbol{\eta} \boldsymbol{\eta}^\top \widetilde{X} \mid \mathcal{D}\big] = \sigma^2 \sum_{i=1}^n \frac{\mathbf{x}^i}{\sqrt{d}} \frac{\mathbf{x}^i}{\sqrt{d}}^\top .$$

Therefore

$$\begin{aligned}
\mathrm{Var}\big(Z(\mathbf{x}) \mid \mathcal{D}, \mathbf{x}\big) &= \frac{1}{\sigma^4} \frac{\mathbf{x}}{\sqrt{d}}^\top \boldsymbol{\Omega} \mathrm{Cov}\big(\widetilde{X}^\top \boldsymbol{\eta} \mid \mathcal{D}\big) \boldsymbol{\Omega}^\top \frac{\mathbf{x}}{\sqrt{d}} \\
&= \frac{1}{\sigma^4} \frac{\mathbf{x}}{\sqrt{d}}^\top \boldsymbol{\Omega} \Big(\sigma^2 \sum_{i=1}^n \frac{\mathbf{x}^i}{\sqrt{d}} \frac{\mathbf{x}^i}{\sqrt{d}}^\top\Big) \boldsymbol{\Omega}^\top \frac{\mathbf{x}}{\sqrt{d}} \\
&= \frac{1}{\sigma^2} \frac{\mathbf{x}}{\sqrt{d}}^\top \boldsymbol{\Omega} \Big(\sum_{i=1}^n \frac{\mathbf{x}^i}{\sqrt{d}} \frac{\mathbf{x}^i}{\sqrt{d}}^\top\Big) \boldsymbol{\Omega}^\top \frac{\mathbf{x}}{\sqrt{d}}.
\end{aligned} \tag{43}$$

Recall that $\widehat{\boldsymbol{\Sigma}} := \frac{1}{n} \sum_{i=1}^n \mathbf{x}^i \mathbf{x}^{i\top}$. Then

$$\sum_{i=1}^n \frac{\mathbf{x}^i}{\sqrt{d}} \frac{\mathbf{x}^i}{\sqrt{d}}^\top = \frac{1}{d} \sum_{i=1}^n \mathbf{x}^i \mathbf{x}^{i\top} = \frac{n}{d} \widehat{\boldsymbol{\Sigma}} = \frac{1}{\alpha} \widehat{\boldsymbol{\Sigma}}, \tag{44}$$

$$\boldsymbol{\Omega} = \sigma^2 \alpha (\widehat{\boldsymbol{\Sigma}} + \hat{R}\mathbf{I})^{-1}.$$

Substituting this into equation 43 yields

$$\mathrm{Var}\big(Z(\mathbf{x}) \mid \mathcal{D}, \mathbf{x}\big) \tag{45}$$

$$= \frac{1}{\sigma^2 \alpha} \frac{\mathbf{x}}{\sqrt{d}}^\top \big[\sigma^2 \alpha (\widehat{\boldsymbol{\Sigma}} + \hat{R}\mathbf{I})^{-1}\big] \widehat{\boldsymbol{\Sigma}} \big[\sigma^2 \alpha (\widehat{\boldsymbol{\Sigma}} + \hat{R}\mathbf{I})^{-1}\big] \frac{\mathbf{x}}{\sqrt{d}}$$

$$= \sigma^2 \alpha \frac{\mathbf{x}}{\sqrt{d}}^\top (\widehat{\boldsymbol{\Sigma}} + \hat{R}\mathbf{I})^{-1} \widehat{\boldsymbol{\Sigma}} (\widehat{\boldsymbol{\Sigma}} + \hat{R}\mathbf{I})^{-1} \frac{\mathbf{x}}{\sqrt{d}}. \tag{46}$$

Let $\mathbf{x} \sim \mathcal{N}(0, \boldsymbol{\Sigma})$ be independent of $\mathcal{D}$, and recall $\frac{\mathbf{x}}{\sqrt{d}} = \mathbf{x}/\sqrt{d}$. For any fixed $d \times d$ matrix $A$,

$$\mathbb{E}_{\mathbf{x}}\Big[\frac{\mathbf{x}}{\sqrt{d}}^\top A \frac{\mathbf{x}}{\sqrt{d}}\Big] = \frac{1}{d} \mathbb{E}_{\mathbf{x}}\big[\mathbf{x}^\top A \mathbf{x}\big] = \frac{1}{d} \mathrm{Tr}(A\boldsymbol{\Sigma}).$$

Applying this to equation 46 gives

$$\begin{aligned}
&\mathbb{E}_{\mathbf{x}}\big[\mathrm{Var}(Z(\mathbf{x}) \mid \mathcal{D}, \mathbf{x})\big] \\
&= \sigma^2 \alpha \frac{1}{d} \mathrm{Tr}\Big((\widehat{\boldsymbol{\Sigma}} + \hat{R}\mathbf{I})^{-1} \widehat{\boldsymbol{\Sigma}} (\widehat{\boldsymbol{\Sigma}} + \hat{R}\mathbf{I})^{-1} \boldsymbol{\Sigma}\Big).
\end{aligned} \tag{47}$$

Since $\mathbb{E}[Z(\mathbf{x}) \mid \mathcal{D}, \mathbf{x}] = 0$, the unconditional variance is

$$\mathrm{Var}\big(Z(\mathbf{x})\big) = \mathbb{E}_{\mathcal{D}, \mathbf{x}}\big[\mathrm{Var}(Z(\mathbf{x}) \mid \mathcal{D}, \mathbf{x})\big], \tag{48}$$

so taking expectation over $\mathcal{D}$ in equation 47 yields

$$\mathrm{Var}\big(Z(\mathbf{x})\big) = \sigma^2 \alpha \, \mathbb{E}_{\mathcal{D}} \left[\frac{1}{d} \mathrm{Tr}\Big((\widehat{\boldsymbol{\Sigma}} + \hat{R}\mathbf{I})^{-1} \widehat{\boldsymbol{\Sigma}} (\widehat{\boldsymbol{\Sigma}} + \hat{R}\mathbf{I})^{-1} \boldsymbol{\Sigma}\Big)\right]. \tag{49}$$

We now invoke the same deterministic–equivalent machinery. One obtains

$$\frac{1}{d} \mathrm{Tr}\Big((\widehat{\boldsymbol{\Sigma}} + \hat{R}\mathbf{I})^{-1} \widehat{\boldsymbol{\Sigma}} (\widehat{\boldsymbol{\Sigma}} + \hat{R}\mathbf{I})^{-1} \boldsymbol{\Sigma}\Big) \xrightarrow[d,n \to \infty]{\mathbb{P}} \frac{\alpha \, m_{\boldsymbol{\Sigma}}^{(2)}(R)}{1 - \alpha \, m_{\boldsymbol{\Sigma}}^{(2)}(R)}, \tag{50}$$

where

$$m_{\boldsymbol{\Sigma}}^{(2)}(R) = \frac{1}{d} \mathrm{Tr}\big[\boldsymbol{\Sigma}^2 (\boldsymbol{\Sigma} + R\mathbf{I})^{-2}\big].$$

Substituting the deterministic equivalent equation 50 into equation 49, we obtain, as $d, n \to \infty$ with $\alpha$ fixed,

$$\begin{aligned}
\mathrm{Var}\big(Z(\mathbf{x})\big) &= \sigma^2 \alpha \frac{1}{d} \mathrm{Tr}\Big((\widehat{\boldsymbol{\Sigma}} + \hat{R}\mathbf{I})^{-1} \widehat{\boldsymbol{\Sigma}} (\widehat{\boldsymbol{\Sigma}} + \hat{R}\mathbf{I})^{-1} \boldsymbol{\Sigma}\Big) \\
&= \sigma^2 \frac{\alpha \, m_{\boldsymbol{\Sigma}}^{(2)}(R)}{1 - \alpha \, m_{\boldsymbol{\Sigma}}^{(2)}(R)}.
\end{aligned} \tag{51}$$

Here we can ignore this contribution as long as

$$\sigma^2 \ll \frac{1 - \alpha \, m_{\boldsymbol{\Sigma}}^{(2)}(R)}{\alpha \, m_{\boldsymbol{\Sigma}}^{(2)}(R)} = \sigma_c(\alpha, R)^2 \tag{52}$$

## C. Proof of Result 2

**Result 10.** *For $T \gg s(\mathbf{x})^2$ the expectation value of the error can be organized as a perturbative series as follows*

$$\begin{aligned}
\delta = \mathbb{E}_{\mathbf{x}}\Big[&\Delta_T(\mathbf{x})^2 + s^2(\mathbf{x}) \\
&+ \Sigma_{l=1}^3 (-1)^l \frac{C_l(\mathbf{x})}{t(\mathbf{x})^l} \prod_{i=1}^l \Big(1 - \frac{i}{k}\Big) + \mathcal{O}\big(t(\mathbf{x})^{-4}\big)\Big]
\end{aligned} \tag{53}$$

*Where we have defined $t(\mathbf{x}) = \frac{T}{2s(\mathbf{x})^2}$ and*

$$C_l(\mathbf{x}) = 2\Delta_T(\mathbf{x})\Delta_R(\mathbf{x}) + s^2(\mathbf{x}) + (l-1)\Delta_R(\mathbf{x})^2 \tag{54}$$

$$\Delta_T(\mathbf{x}) := m(\mathbf{x}) - \mu_T(\mathbf{x}), \quad \Delta_R(\mathbf{x}) := m(\mathbf{x}) - \mu_R(\mathbf{x}). \tag{55}$$

*All other quantities are as in Result 1.*

*Proof-sketch.* Consider a random variable $x$ and we are interested in concentration properties of the function $f(x) = \log x$. Here we will discuss a controlled approximation

technique to evaluate the expectation value of $f$. Simplest approach is to Taylor expand $f$ around the expectation value of $x$ denoted by $\bar{x}$

$$
\begin{aligned}
f(x) &= \log \bar{x} + \frac{1}{\bar{x}}(x - \bar{x}) - \frac{1}{2\bar{x}^2}(x - \bar{x})^2 \\
&\quad + \frac{1}{3\bar{x}^3}(x - \bar{x})^3 + \mathcal{O}\left(\frac{(x - \bar{x})^4}{\bar{x}^4}\right)
\end{aligned}
\tag{56}
$$

Taking expectation value of both sides of the equation above we get the following expression

$$
\begin{aligned}
\mathbb{E}(\log x) &= \log \mathbb{E}(x) - \frac{\mathbb{E}(x^2) - \mathbb{E}(x)^2}{2\mathbb{E}(x)^2} \\
&\quad + \frac{\mathbb{E}(x^3) - 3\mathbb{E}(x^2)\mathbb{E}(x) + 2\mathbb{E}(x)^3}{3\mathbb{E}(x)^3} \\
&\quad + \mathcal{O}\left(\frac{\mathbb{E}((x - \mathbb{E}(x))^4)}{\mathbb{E}(x)^4}\right)
\end{aligned}
\tag{57}
$$

This approximation scheme is only useful when higher order corrections are relatively small. Next we use this to derive the result stated in the Result above.

It will be convenient to define partition function density given by

$$
z(\mathbf{J}, \mathbf{p}) = \frac{1}{k}\sum_{i=1}^{k} e^{-\frac{1}{T}E_{\mathbf{w}_R}(p_i) - J_i E_{\mathbf{w}_T}(p_i)}, \quad E_{\mathbf{w}}(p) = (p - \mathbf{w} \cdot \frac{\mathbf{x}}{\sqrt{d}})^2
\tag{58}
$$

The expectation value of $z(\mathbf{J}, \mathbf{p})$ when $p_i$ is sampled from the following distribution

$$
p_i \sim \mathcal{N}(m, \Sigma = s^2), \quad i = 1, 2, \ldots, k
\tag{59}
$$

gives disorder averaged, over $k$ samples, partition function, with an additional chemical potential $\mathbf{J}$, of free particles at temperature $T$. The error given in (7) can be expressed in terms of this quantity as follows

$$
\begin{aligned}
\delta &= \mathbb{E}_{\mathbf{p}}\left(\frac{\sum_{i=1}^{k} E_{\mathbf{w}_T}(p_i) e^{-\frac{1}{T}E_{\mathbf{w}_R}(p_i) - J_i E_{\mathbf{w}_T}(p_i)}}{\sum_{i=1}^{k} e^{-\frac{1}{T}E_{\mathbf{w}_R}(p_i) - J_i E_{\mathbf{w}_T}(p_i)}}\right) \\
&= -\mathbb{E}_{\mathbf{p}}\sum_i \partial_{J_i}(\log z(\mathbf{J}, \mathbf{p}))|_{\mathbf{J}=0} \\
&\approx -\sum_i \partial_{J_i}\left(\log \mathbb{E}_{\mathbf{p}}(z(\mathbf{J}, \mathbf{p})) - \frac{\mathbb{E}_{\mathbf{p}}(z(\mathbf{J}, \mathbf{p})^2) - \mathbb{E}_{\mathbf{p}}(z(\mathbf{J}, \mathbf{p}))^2}{2\mathbb{E}_{\mathbf{p}}(z(\mathbf{J}, \mathbf{p}))^2}\right. \\
&\quad \left. + \frac{\mathbb{E}_{\mathbf{p}}(z(\mathbf{J}, \mathbf{p})^3) - 3\mathbb{E}_{\mathbf{p}}(z(\mathbf{J}, \mathbf{p})^2)\mathbb{E}_{\mathbf{p}}(z(\mathbf{J}, \mathbf{p})) + 2\mathbb{E}_{\mathbf{p}}(z(\mathbf{J}, \mathbf{p}))^3}{3\mathbb{E}_{\mathbf{p}}(z(\mathbf{J}, \mathbf{p}))^3}\right)
\end{aligned}
\tag{60}
$$

To go from the second to the third line we have made an approximation, we will present the domain of validity of the approximation shortly. To this end we compute,

$$
\mathbb{E}_{\mathbf{p}}(z(\mathbf{J}, \mathbf{p})) = \frac{1}{k}\sum_{i=1}^{k}\mathbb{E}_p(e^{-\frac{1}{T}E_{\mathbf{w}_R}(p) - J_i E_{\mathbf{w}_T}(p)})
\tag{61}
$$

The required expectation values can be expressed in terms of the following function

$$
\begin{aligned}
&h\big(m_1, m_2, m_3;\, s_1^2, s_2^2, s_3^2\big) \\
&= \frac{\exp\left[-\dfrac{\displaystyle\sum_{1 \leq i \neq j \neq k \leq 3}\left(\frac{1}{2}m_i^2(s_j^2 + s_k^2) - m_i m_j s_k^2\right)}{2\prod_{i=1}^{3}s_i^2\displaystyle\sum_{i=1}^{3}\frac{1}{s_i^2}}\right]}{2\pi\sqrt{\prod_{i=1}^{3}s_i^2\displaystyle\sum_{i=1}^{3}\frac{1}{s_i^2}}}.
\end{aligned}
\tag{62}
$$

The moments of $z(\mathbf{J}, \mathbf{p})$ are given by

$$
\begin{aligned}
&\mathbb{E}_{\mathbf{p}}\big[z(\mathbf{J}, \mathbf{p})\big] \\
&= \frac{1}{k}\sum_{i=1}^{k}\pi\sqrt{\frac{T}{J_i}}\, h\Big(m, \frac{\mathbf{w}_R \cdot \mathbf{x}}{\sqrt{d}}, \frac{\mathbf{w}_T \cdot \mathbf{x}}{\sqrt{d}}; s^2, \frac{T}{2}, \frac{1}{2J_i}\Big),
\end{aligned}
\tag{63}
$$

$$
\begin{aligned}
\mathbb{E}_{\mathbf{p}}\big[z(\mathbf{J}, \mathbf{p})^2\big] &= \frac{1}{k^2}\left[\sum_{\substack{i,j=1 \\ i \neq j}}^{k}\pi\sqrt{\frac{T}{J_i}}\, h\Big(m, \frac{\mathbf{w}_R \cdot \mathbf{x}}{\sqrt{d}}, \frac{\mathbf{w}_T \cdot \mathbf{x}}{\sqrt{d}}; s^2, \frac{T}{2}, \frac{1}{2J_i}\Big)\right. \\
&\quad \times \pi\sqrt{\frac{T}{J_j}}\, h\Big(m, \frac{\mathbf{w}_R \cdot \mathbf{x}}{\sqrt{d}}, \frac{\mathbf{w}_T \cdot \mathbf{x}}{\sqrt{d}}; s^2, \frac{T}{2}, \frac{1}{2J_j}\Big) \\
&\quad \left. + \sum_{i=1}^{k}\pi\sqrt{\frac{T}{4J_i}}\, h\Big(m, \frac{\mathbf{w}_R \cdot \mathbf{x}}{\sqrt{d}}, \frac{\mathbf{w}_T \cdot \mathbf{x}}{\sqrt{d}}; s^2, \frac{T}{4}, \frac{1}{4J_i}\Big)\right],
\end{aligned}
$$

$$
\begin{aligned}
&\mathbb{E}_{\mathbf{p}}\big[z(\mathbf{J}, \mathbf{p})^3\big] \\
&= \frac{1}{k^3}\left[\sum_{1 \leq i < j < \ell \leq k}6\pi\sqrt{\frac{T}{J_i}}\, h\Big(m, \frac{\mathbf{w}_R \cdot \mathbf{x}}{\sqrt{d}}, \frac{\mathbf{w}_T \cdot \mathbf{x}}{\sqrt{d}}; s^2, \frac{T}{2}, \frac{1}{2J_i}\Big)\right. \\
&\quad \times \pi\sqrt{\frac{T}{J_j}}\, h\Big(m, \frac{\mathbf{w}_R \cdot \mathbf{x}}{\sqrt{d}}, \frac{\mathbf{w}_T \cdot \mathbf{x}}{\sqrt{d}}; s^2, \frac{T}{2}, \frac{1}{2J_j}\Big) \\
&\quad \times \pi\sqrt{\frac{T}{J_\ell}}\, h\Big(m, \frac{\mathbf{w}_R \cdot \mathbf{x}}{\sqrt{d}}, \frac{\mathbf{w}_T \cdot \mathbf{x}}{\sqrt{d}}; s^2, \frac{T}{2}, \frac{1}{2J_\ell}\Big) \\
&\quad + \sum_{\substack{i,j=1 \\ i \neq j}}^{k}3\pi\sqrt{\frac{T}{4J_i}}\, h\Big(m, \frac{\mathbf{w}_R \cdot \mathbf{x}}{\sqrt{d}}, \frac{\mathbf{w}_T \cdot \mathbf{x}}{\sqrt{d}}; s^2, \frac{T}{4}, \frac{1}{4J_i}\Big) \\
&\quad \times \pi\sqrt{\frac{T}{J_j}}\, h\Big(m, \frac{\mathbf{w}_R \cdot \mathbf{x}}{\sqrt{d}}, \frac{\mathbf{w}_T \cdot \mathbf{x}}{\sqrt{d}}; s^2, \frac{T}{2}, \frac{1}{2J_j}\Big) \\
&\quad \left. + \sum_{i=1}^{k}\pi\sqrt{\frac{T}{9J_i}}\, h\Big(m, \frac{\mathbf{w}_R \cdot \mathbf{x}}{\sqrt{d}}, \frac{\mathbf{w}_T \cdot \mathbf{x}}{\sqrt{d}}; s^2, \frac{T}{6}, \frac{1}{6J_i}\Big)\right].
\end{aligned}
$$

This leads to the following perturbative expansion in terms

of $t = \frac{T}{2s^2}$:

$$\delta(\mathbf{x}) = (m - \mathbf{w}_T \cdot \frac{\mathbf{x}}{\sqrt{d}})^2 + s^2$$
$$- \frac{1}{t}\left(1 - \frac{1}{k}\right)\left((m - \frac{\mathbf{w}_R \cdot \mathbf{x}}{\sqrt{d}})\left(2(m - \frac{\mathbf{w}_T \cdot \mathbf{x}}{\sqrt{d}})\right)\right) + s^2$$
$$+ \frac{1}{t^2}\left(1 - \frac{1}{k}\right)\left(1 - \frac{2}{k}\right)$$
$$\times \left(\left((m - \frac{\mathbf{w}_R \cdot \mathbf{x}}{\sqrt{d}})\left(2\left(m - \frac{\mathbf{w}_T \cdot \mathbf{x}}{\sqrt{d}}\right)\right.\right.$$
$$+ \left(m - \frac{\mathbf{w}_R \cdot \mathbf{x}}{\sqrt{d}}\right)\right) + s^2\right) + O\left(\frac{1}{t^3}\right) \tag{64}$$

Repeating this technique to higher order is straight forward but algebraically tedious. Here we quote the result to one higher order

$$\delta(\mathbf{x}) = (m - \mathbf{w}_T \cdot \frac{\mathbf{x}}{\sqrt{d}})^2 + s^2$$
$$+ \Sigma_{l=1}^{3}(-1)^l\left(1 - \frac{1}{k}\right)\left(1 - \frac{2}{k}\right)\ldots\left(1 - \frac{l}{k}\right)\frac{1}{t^l}C_l(\mathbf{x})$$
$$+ O\left(\frac{1}{t^4}\right)$$
$$C_l(\mathbf{x}) = 2\left(m - \frac{\mathbf{w}_R \cdot \mathbf{x}}{\sqrt{d}}\right)(m - \frac{\mathbf{w}_T \cdot \mathbf{x}}{\sqrt{d}})$$
$$+ s^2(\mathbf{x}) + (l - 1)(m - \frac{\mathbf{w}_R \cdot \mathbf{x}}{\sqrt{d}})^2 \tag{65}$$

# D. Proof of Result 3

**Result 11** (Low-$T$ best-of-$k$ sampling). *When we have access to the exact teacher weight $\mathbf{w}_R = \mathbf{w}_T = \mathbf{w}$, the leading order result for $T \to 0$ followed by $k \to \infty$ is given by*

$$\delta(\mathbf{x}) = \frac{\pi}{k^2} s^2(\mathbf{x}) \exp\left(\frac{\Delta_T(\mathbf{x})^2}{s^2(\mathbf{x})}\right) \tag{66}$$

*All the quantities are as in Result 1 and Result 2.*

*Proof-sketch.* In $T \to 0$ limit it is natural to approximate the softmax by the sample with highest reward, i.e.,

$$\delta(\mathbf{x}) = \mathbb{E}_{y_i \sim \mathcal{N}(m,\Sigma), i=1,\ldots,k}\left(\left(y_k' - \mathbf{w}_T \cdot \frac{\mathbf{x}}{\sqrt{d}}\right)^2 : \right.$$
$$\left. y_k' = \arg\min_{y \in (y_1, y_2, \ldots, y_k)}\left(y - \mathbf{w}_R \cdot \frac{\mathbf{x}}{\sqrt{d}}\right)^2\right) \tag{67}$$

First note that the distribution of the penalty is a non-central chi-squared distribution with one degree of freedom:

$$-v \equiv \frac{1}{s^2}\left(y - \mathbf{w}_R \cdot \frac{\mathbf{x}}{\sqrt{d}}\right)^2 \sim \chi_1^2(\lambda), \lambda = \frac{(m - \mathbf{w}_R \cdot \frac{\mathbf{x}}{\sqrt{d}})^2}{s^2} \tag{68}$$

We focus on the situation of perfect reward $\mathbf{w}_R = \mathbf{w}_T$ in the high reward regime. In this case,

$$\delta(\mathbf{x}) = s^2\mathbb{E}(-v_{max}), \quad v_{max} = \max_{-v_i \sim \chi_1^2(\lambda)}(v_1, v_2, \ldots, v_k) \tag{69}$$

Since we are looking for the minimum of the chi-squared distributed variables the extreme value statistics is more involved and the final distribution is different compared to the analysis for maximum.

Next we focus on $k \to \infty$ limit for analytical tractability. Probability distribution function $\varphi$ and cumulative distribution function $F$ of $v \leq 0$ is given by

$$\varphi(v) = \frac{e^{\frac{v-\lambda}{2}}\cos\left(\sqrt{\lambda v}\right)}{\sqrt{2\pi}\sqrt{-v}},$$
$$F(v) = 1 - \frac{1}{2}\left(\mathrm{erf}\left(\frac{\sqrt{-v} - \sqrt{\lambda}}{\sqrt{2}}\right) + \mathrm{erf}\left(\frac{\sqrt{\lambda} + \sqrt{-v}}{\sqrt{2}}\right)\right) \tag{70}$$

Note that as $k \to \infty$ the degenerate distribution concentrates near $v_F = 0$. Given that $v_F$ is finite, the extreme distribution could be either of Weibull or Gumbel type. Next we show that it is not Gumbel type. To see this we calculate the auxiliary function for the Gumbel type using

$$a(v) = \frac{1 - F(v)}{F'(v)}, \quad \lim_{v \to v_F} a'(v) \neq 0 \tag{71}$$

This limit on right ensures it cannot be of Gumbel type. To identify the Weibull distribution

$$\Psi_\alpha(x) = e^{-(-x)^\alpha} \quad \text{for } x \leq 0, \quad 1 \quad \text{otherwise} \tag{72}$$

We calculate

$$\alpha = \lim_{v \to v_F} \frac{(v_F - v)F'(v)}{1 - F(x)} = \frac{1}{2} \tag{73}$$

This ensures the cumulative distribution function of $(v_{max} - d_k)/c_k$ is $\Psi_\alpha$ where

$$d_k = v_F = 0$$
$$c_k = v_F - F^{\leftarrow}\left(1 - \frac{1}{k}\right) = \frac{\pi}{2k^2}e^\lambda \tag{74}$$

The probability density for $-v_{max}/c_k$ is

$$p_{max}(-v_{max}/c_k) = \alpha(-v_{max}/c_k)^{\alpha-1}e^{-(-v_{max}/c_k)^\alpha}, \quad v_{max} \leq 0 \tag{75}$$

This gives the following expression for the error in $t \to 0, k \to \infty$ limit (taken in this order)

$$\delta(\mathbf{x}) = s^2 c_k \mathbb{E}(-v_{max}/c_k)$$
$$= s^2 c_k \Gamma(1 + 1/\alpha)$$
$$= \frac{\pi s^2}{k^2}e^{\frac{(m - \mathbf{w} \cdot \frac{\mathbf{x}}{\sqrt{d}})^2}{s^2}} \tag{76}$$

# E. Proof of Remark 6

*Remark* 8. Given access to exact teacher weight, it is beneficial to scale inference-time compute over adding more training samples in the following regime: consider $T \to 0$ followed by $k \to \infty$ with

$$\frac{\gamma^2}{d}\mathrm{Tr}(\mathbf{B}_R\mathbf{\Sigma}) \ll \sigma^2, \quad R \ll \sigma^2 \tag{77}$$

That is within this domain,

$$\frac{\partial \log \delta}{\partial \log k} = -2, \qquad \frac{\partial \log \delta}{\partial \log n} = -\frac{\alpha\partial_\alpha(\mathbf{u}^\top\mathbf{\Sigma}\mathbf{u})}{\sigma^2 d - 2\mathbf{u}^\top\mathbf{\Sigma}\mathbf{u}} \tag{78}$$

and hence

$$\left|\frac{\partial \log \delta}{\partial \log k}\right| \gg \left|\frac{\partial \log \delta}{\partial \log n}\right| \tag{79}$$

*Proof-sketch.* Our goal is to put upper bound on the magnitude of the derivative w.r.t. $n$. We proceed systematically by working in the eigen basis of sample variance. By the spectral Theorem for real symmetric matrices, there exists an orthogonal matrix $Q \in O(d)$ and a diagonal $\Lambda = \mathrm{diag}(\lambda_1, \ldots, \lambda_d)$ with $\lambda_i \geq 0$ such that

$$\mathbf{\Sigma} = Q\Lambda Q^\top, \qquad \mathbf{B}_R = Q\,\mathrm{diag}\Big(\frac{R}{\lambda_i + R}\Big)Q^\top.$$

This immediately gives following inequalities

$$m_{\mathbf{\Sigma}}(R) = \frac{1}{d}\mathrm{Tr}\big[\mathbf{\Sigma}(\mathbf{\Sigma}+R\mathbf{I})^{-1}\big] = \frac{1}{d}\sum_{i=1}^d \frac{\lambda_i}{\lambda_i + R} < 1 \tag{80}$$

$$m'_{\mathbf{\Sigma}}(R) = -\frac{1}{d}\mathrm{Tr}\big[\mathbf{\Sigma}(\mathbf{\Sigma}+R\mathbf{I})^{-2}\big] = -\frac{1}{d}\sum_{i=1}^d \frac{\lambda_i}{(\lambda_i + R)^2} \leq 0 \tag{81}$$

We are interested in putting upper bound on the following quantity

$$\alpha\,\partial_\alpha(\mathbf{u}^\top\mathbf{\Sigma}\mathbf{u}) = \alpha\,F'(R)\,\frac{dR}{d\alpha},$$

$$F(R) = \mathbf{u}^\top\mathbf{\Sigma}\mathbf{u} = \sum_{i=1}^d \frac{R^2\lambda_i}{(\lambda_i + R)^2}\,\tilde{w}_i^2, \quad \tilde{\mathbf{w}} = Q^\top\mathbf{w} \tag{82}$$

We will first derive an upper bound on $F'(R)$ and an upper bound on $\frac{dR}{d\alpha}$. To this goal we proceed by noting that $\max_{\lambda \geq 0}\frac{\lambda^2}{(\lambda+R)^3} = \frac{4}{27}\frac{1}{R}$ (attained at $\lambda = 2R$) and we have

$$F'(R) = \sum_{i=1}^d \frac{2R\lambda_i^2}{(\lambda_i + R)^3}\,\tilde{w}_i^2 \leq 2R\cdot\frac{4}{27}\frac{1}{R}\,\|\mathbf{w}\|^2 = \frac{8}{27}\,d.$$

Next we focus on the deterministic equivalents equation

$$\hat{R} = R\big(1 - \alpha m_{\mathbf{\Sigma}}(R)\big), \qquad \hat{R} = \frac{\sigma^2\alpha}{\gamma^2}.$$

Differentiate both sides w.r.t. $\alpha$:

$$\Big((1 - \alpha m_{\mathbf{\Sigma}}) - \alpha R m'_{\mathbf{\Sigma}}\Big) R' = \frac{\sigma^2}{\gamma^2} + R\,m_{\mathbf{\Sigma}}(R).$$

Using the inequalities in equation (80)

$$R' \leq \frac{\frac{\sigma^2}{\gamma^2} + R\,m_{\mathbf{\Sigma}}(R)}{\hat{R}/R} \leq \frac{\frac{\sigma^2}{\gamma^2} + R}{\hat{R}/R} = \frac{\sigma^2}{\gamma^2}\frac{R}{\hat{R}} + \frac{R^2}{\hat{R}}.$$

This gives us the desired upper bound

$$\alpha\,\partial_\alpha(\mathbf{u}^\top\mathbf{\Sigma}\,\mathbf{u}) = \alpha\,F'(R)\,\frac{dR}{d\alpha} \leq \frac{8}{27}\,d\Big(R + \frac{\gamma^2 R^2}{\sigma^2}\Big).$$

Finally we aim for establishing a lower bound on $\sigma^2 d - 2\,\mathbf{u}^\top\mathbf{\Sigma}\,\mathbf{u}$. This is achieved by the following observation

$$\frac{R^2\lambda}{(\lambda + R)^2} \leq \frac{R^2(\lambda + R)}{(\lambda + R)^2} = \frac{R^2}{\lambda + R} \leq R,$$

Hence

$$\mathbf{u}^\top\mathbf{\Sigma}\,\mathbf{u} = \sum_{i=1}^d \frac{R^2\lambda_i}{(\lambda_i + R)^2}\,\tilde{w}_i^2 \leq R\sum_{i=1}^d \tilde{w}_i^2 = R\,\|\mathbf{w}\|^2 = R\,d.$$

Therefore

$$\sigma^2 d - 2\,\mathbf{u}^\top\mathbf{\Sigma}\,\mathbf{u} \geq d\,(\sigma^2 - 2R).$$

Putting both the results together, when $R/\sigma^2 \ll 1$ we get

$$\left|\frac{\partial \log \delta}{\partial \log n}\right| \leq \frac{\frac{8}{27}\,d\,(R + \gamma^2 R^2/\sigma^2)}{d(\sigma^2 - 2R)} = \frac{8}{27}\cdot\frac{R}{\sigma^2}\cdot\frac{1 + \gamma^2 R/\sigma^2}{1 - 2R/\sigma^2} \ll 1$$

# F. Inference time scaling of generic probabilistic models

Let $p_{\mathbf{x}}(y) := p(y \mid \mathbf{x}, \mathcal{D})$ be an arbitrary predictive distribution for a fixed input $\mathbf{x}$, let $r(y, \mathbf{x})$ be an arbitrary reward, and let $\ell(y, \mathbf{x})$ be the task loss we care about. Define the finite-sample reranked loss

$$\delta_k^{(\ell,r)}(\mathbf{x}, T) := \mathbb{E}_{Y_{1:k}\sim p_{\mathbf{x}}}\left[\frac{\sum_{i=1}^k \ell(Y_i, \mathbf{x})e^{r(Y_i,\mathbf{x})/T}}{\sum_{j=1}^k e^{r(Y_j,\mathbf{x})/T}}\right]. \tag{83}$$

*(a) High-temperature regime.*

Expanding in $\varepsilon := 1/T$ gives $e^{\varepsilon r} = 1 + \varepsilon r + \frac{\varepsilon^2}{2}r^2 + O(\varepsilon^3)$, and a direct expansion yields

$$\begin{aligned}\delta_k^{(\ell,r)}(\mathbf{x}, T) =\ & \mu_\ell(\mathbf{x}) + \frac{k-1}{k}\frac{1}{T}\,\mathrm{Cov}_x(\ell, r) \\ & + \frac{(k-1)(k-2)}{2k^2}\frac{1}{T^2}\mathrm{Cov}_x\big(\ell, (r - \mu_r(\mathbf{x}))^2\big) \\ & + O(T^{-3})\end{aligned} \tag{84}$$

where $\mu_\ell(\mathbf{x}) := \mathbb{E}_{p_\mathbf{x}}[\ell(Y, \mathbf{x})], \mu_r(\mathbf{x}) := \mathbb{E}_{p_\mathbf{x}}[r(Y, \mathbf{x})]$, and $\mathrm{Cov}_x$ denotes covariance under $Y \sim p_\mathbf{x}$.

*First-order term.* The leading correction is $\frac{k-1}{k} \frac{1}{T} \mathrm{Cov}_x(\ell, r)$. Hence, to first order, reranking is beneficial precisely when $\mathrm{Cov}_x(\ell, r) < 0$, that is, when larger reward is correlated with smaller task loss under the base predictive distribution. This conclusion is completely general. Moreover, at this order:

- the dependence on $k$ is monotone, since it enters only through $(k-1)/k$;

- the dependence on $T$ is monotone, since the correction scales as $1/T$;

- therefore neither an optimal temperature nor a finite optimal $k$ can appear at first order alone.

In the asymptotic high-$T$ regime where the first-order term dominates, only the basic alignment effect survives.

*Second-order term.* The next coefficient is $\frac{(k-1)(k-2)}{2k^2} \frac{1}{T^2} \mathrm{Cov}_x(\ell, (r - \mu_r(\mathbf{x}))^2)$. This term is still completely general, this term measures how the fluctuation of the reward correlates with the task loss. More explicitly:

- If $\mathrm{Cov}_x(\ell, (r - \mu_r)^2) > 0$, then higher-loss draws tend to occur where the reward fluctuates more strongly around its mean. In that case, aggressive reranking can over-amplify reward noise or reward misspecification, which is exactly the mechanism that can generate a finite optimal temperature or a finite optimal sample budget.

- If $\mathrm{Cov}_x(\ell, (r - \mu_r)^2) < 0$, then lower-loss draws tend to come with larger reward spread. In that case, the second-order correction helps the selector distinguish good draws more strongly, and reranking remains helpful.

The expansion equation 84 does not use Gaussianity. It only requires the relevant moments to exist under $p_\mathbf{x}$. So in the high-temperature regime, what Gaussianity gives us is an analytically tractable closed form for those moments and can be easily generalized to heavy-tailed or multi-modal distributions.

The main text in the paper corresponds to the following choice $\ell(y, \mathbf{x}) = (y - \mu_T(\mathbf{x}))^2, r(y, \mathbf{x}) = -(y - \mu_R(\mathbf{x}))^2, p_\mathbf{x} = f_x := \mathcal{N}(m(\mathbf{x}), s(\mathbf{x})^2)$. This choice allows us to explicitly perform calculation of the quantities appearing above.

*(b) Low-temperature regime.*

Here the key object is not the global shape of the predictive distribution, but its local behavior near the reward maximizer. If the judge-induced score $r(y, \mathbf{x})$ is smooth and has a local maximizer $y_\star(\mathbf{x})$, then

$$r(y, \mathbf{x}) = r(y_\star, \mathbf{x}) - \frac{\kappa(\mathbf{x})}{2}(y - y_\star(\mathbf{x}))^2 + O(|y - y_\star(\mathbf{x})|^3),$$
(85)

with $\kappa(\mathbf{x}) := -\partial_y^2 r(y_\star, \mathbf{x}) > 0$. Thus, locally, the reward-weighted selector behaves like our quadratic model with

$$\mu_R(\mathbf{x}) = y_\star(\mathbf{x}).$$
(86)

The additive constant $r(y_\star, \mathbf{x})$ drops out of the softmax, while $\kappa(\mathbf{x})$ simply rescales $T$. Hence, when $T$ is small, only samples very close to $\mu_R(\mathbf{x})$ matter, so the quadratic approximation is the correct universal local model. We therefore focus on $T = 0$.

We will find following definition useful for $Y_i \overset{\text{iid}}{\sim} p_\mathbf{x}$

$$Y_{(R)}(\mathbf{x}) := \arg \min_{1 \leq i \leq k} |Y_i - \mu_R(\mathbf{x})|, S_k(\mathbf{x}) := Y_{(R)}(\mathbf{x}) - \mu_R(\mathbf{x}).$$
(87)

Define the local density around the reward target by

$$g_{R,\mathbf{x}}(u) := p_\mathbf{x}(\mu_R(\mathbf{x}) + u).$$
(88)

Assume that

$$g_{R,\mathbf{x}}(u) = c_{R,\mathbf{x}}|u|^\beta + d_{R,\mathbf{x}}u|u|^\beta + o(|u|^{\beta+1}), \qquad u \to 0.$$
(89)

Then 'coverage' is given by

$$H_{R,\mathbf{x}}(r) := \Pr(|Y - \mu_R(\mathbf{x})| \leq r \mid \mathbf{x}) = \int_{-r}^{r} g_{R,\mathbf{x}}(u)\, du$$

$$= 2c_{R,\mathbf{x}} \int_0^r u^\beta\, du + o(r^{\beta+1})$$

$$=: A_{R,\mathbf{x}} r^{\beta+1} + o(r^{\beta+1}), \qquad A_{R,\mathbf{x}} := \frac{2c_{R,\mathbf{x}}}{\beta+1}.$$
(90)

Therefore for small $r$

$$\Pr(|S_k(\mathbf{x})| > r \mid \mathbf{x}) = (1 - H_{R,\mathbf{x}}(r))^k \approx \exp(-kA_{R,\mathbf{x}} r^{\beta+1}),$$
(91)

and integration by parts gives

$$\mathbb{E}[S_k(\mathbf{x})^2 \mid \mathbf{x}] = 2 \int_0^\infty r \Pr(|S_k(\mathbf{x})| > r \mid \mathbf{x})\, dr$$

$$\approx 2 \int_0^\infty r\, e^{-kA_{R,\mathbf{x}} r^{\beta+1}}\, dr$$
(92)

$$= \frac{2}{\beta+1} A_{R,\mathbf{x}}^{-\frac{2}{\beta+1}} \Gamma\left(\frac{2}{\beta+1}\right) k^{-\frac{2}{\beta+1}}.$$

Here we used the fact that for large $k$, significant contribution comes from small $r$ region.

When $\mu_R(\mathbf{x}) = \mu_T(\mathbf{x})$, this is exactly the generalization error:

$$\mathbb{E}\big[(Y_{(R)} - \mu_T(\mathbf{x}))^2 \mid \mathbf{x}\big] \asymp k^{-\frac{2}{\beta+1}}.$$

Hence in this case the decay of the generalization error is governed by the 'coverage' near the target. For the Gaussian case in the main text, $\beta = 0$, so we recover the $k^{-2}$ law.

Since ties occur with probability zero under a continuous density, the density of $S_k$ is

$$h_{k,\mathbf{x}}(u) = k\, g_{R,\mathbf{x}}(u) \big(1 - H_{R,\mathbf{x}}(|u|)\big)^{k-1}. \tag{93}$$

Hence

$$\mathbb{E}[S_k \mid \mathbf{x}] = k \int_{\mathbb{R}} u\, g_{R,\mathbf{x}}(u)\big(1 - H_{R,\mathbf{x}}(|u|)\big)^{k-1} du$$

$$= k \int_0^\infty u\big(g_{R,\mathbf{x}}(u) - g_{R,\mathbf{x}}(-u)\big)$$
$$\big(1 - H_{R,\mathbf{x}}(u)\big)^{k-1} du. \tag{94}$$

$$\approx 2 d_{R,\mathbf{x}}\, k \int_0^\infty u^{\beta+2} e^{-k A_{R,\mathbf{x}} u^{\beta+1}}\, du$$

$$= \frac{2 d_{R,\mathbf{x}}}{\beta+1} A_{R,\mathbf{x}}^{-\frac{\beta+3}{\beta+1}} \Gamma\left(\frac{\beta+3}{\beta+1}\right) k^{-\frac{2}{\beta+1}}.$$

Finally, writing

$$\Delta_x := \mu_R(\mathbf{x}) - \mu_T(\mathbf{x}),$$

the generalization error in the mismatched case is

$$\mathbb{E}\big[(Y_{(R)} - \mu_T(\mathbf{x}))^2 \mid \mathbf{x}\big]$$
$$= \Delta_x^2 + 2\Delta_x \mathbb{E}[S_k \mid \mathbf{x}] + \mathbb{E}[S_k^2 \mid \mathbf{x}]$$
$$= \Delta_x^2 + 4\frac{\Delta_x d_{R,\mathbf{x}}}{\beta+1} A_{R,\mathbf{x}}^{-\frac{\beta+3}{\beta+1}} \Gamma\left(\frac{\beta+3}{\beta+1}\right) k^{-\frac{2}{\beta+1}} \tag{95}$$
$$+ \frac{2}{\beta+1} A_{R,\mathbf{x}}^{-\frac{2}{\beta+1}} \Gamma\left(\frac{2}{\beta+1}\right) k^{-\frac{2}{\beta+1}}.$$

So when reward and target are misaligned, the error approaches the nonzero floor $\Delta_x^2$, with the scaling exponent $2/(\beta+1)$ controlling the first correction. The graph in Figure 9 can be fitted into $\beta > 0$.

In particular, for settings in the main text $\beta = 0$. Consequently, at large $k$

$$\delta(\mathbf{x}) = \mathbb{E}\big[(Y_{(R)} - \mu_T)^2\big]$$
$$= (\mu_R(\mathbf{x}) - \mu_T(\mathbf{x}))^2$$
$$+ (\mu_R(\mathbf{x}) - \mu_T(\mathbf{x})) \frac{f_x'\big(\mu_R(\mathbf{x})\big)}{f_x\big(\mu_R(\mathbf{x})\big)^3} \frac{1}{k^2} \tag{96}$$
$$+ \frac{1}{2k^2 f_x\big(\mu_R(\mathbf{x})\big)^2}.$$

In particular, note that the second term could be positive or negative depending on the level of reward misspecification.

We have presented numerical results for $T = 0$ misspecified reward in Appendix H.1 after averaging over $\mathbf{x}$. We see that these general theoretical expectations are fulfilled.

**Connection to tilted policy learning.** The large $k$ limit that we study here is connected to learning an exponentially tilted policy relative to base generative model as we explain next. The reward-reweighted exponentially tilted policy is (Aminian et al., 2025; Faria and Smith, 2025)

$$\pi_T(y \mid \mathbf{x}, \mathcal{D}) = \frac{e^{r(y,\mathbf{x})/T} p_x(y)}{Z_T(\mathbf{x})},$$
$$Z_T(\mathbf{x}) := \int e^{r(y',\mathbf{x})/T} p_x(y') dy'. \tag{97}$$

Now consider the finite-sample procedure in our paper: given the sampled set $Y_{1:k}$, this induces the empirical soft policy

$$\widehat{\pi}_{k,T}(y \mid \mathbf{x}, \mathcal{D}; Y_{1:k}) := \sum_{i=1}^k \frac{e^{r(Y_i,\mathbf{x})/T}}{\sum_{j=1}^k e^{r(Y_j,\mathbf{x})/T}} \delta_{Y_i}(y). \tag{98}$$

Therefore, for any test function $\varphi$,

$$\widehat{\pi}_{k,T}(\varphi) = \int \varphi(y)\, \widehat{\pi}_{k,T}(y) dy = \frac{\sum_{i=1}^k \varphi(Y_i) e^{r(Y_i,\mathbf{x})/T}}{\sum_{j=1}^k e^{r(Y_j,\mathbf{x})/T}}.$$

As $k \to \infty$, $\widehat{\pi}_{k,T}(\varphi) \to \pi_T(\varphi)$. If $\mathbb{E}$ denotes expectation over the inference-time samples $Y_{1:k}$, the generalization error is given by, $\delta(\mathbf{x}) = \mathbb{E}(\widehat{\pi}_{k,T}(\ell))$ with $\ell(y) := (y - \mu_T(\mathbf{x}))^2$. So the finite-$k$ rule studied in our paper is a finite sampled approximation to the reward-reweighted exponentially tilted policy, and our theorem computes the leading corrections to generalization error in a closed form under this policy.

*(i) A reward–loss coupling coefficient controlling expected finite sample policy difference.*

A natural measure of how strongly reward and task are coupled under the base predictive law $p_x$ is

$$\mathcal{A}(\mathbf{x}) := -\mathrm{Cov}_{\mathbf{x}}(\ell, r).$$

Indeed, the large-$T$ expansion discussed previously gives

$$\mathbb{E}(\widehat{\pi}_{k,T}(\ell)) - \pi_T(\ell) = \frac{1}{kT} \mathcal{A}(\mathbf{x}) + O(T^{-2}). \tag{99}$$

So the leading gain from reranking is controlled exactly by this covariance-type coupling parameter.

*(ii) A decoupling coefficient controlling random finite sample policy difference.*

If instead one wants the quantity $\widehat{\pi}_{k,T}(\ell) - \pi_T(\ell)$ that measures how hard it is for a finite sample budget $k$ to realize

the ideal tilted policy $\pi_T$, the natural continuous object is the second moment of the importance weights:

$$C(\mathbf{x}) := \frac{\mathbb{E}_{Y \sim p_x}\left[(e^{r(Y,\mathbf{x})/T})^2\right]}{\left(\mathbb{E}_{Y \sim p_x}\left[e^{r(Y,\mathbf{x})/T}\right]\right)^2}. \tag{100}$$

If the weights are well spread out, then the estimator behaves like an average of roughly $k$ useful samples, and therefore its error has the usual scale $k^{-1/2}$. By contrast, if the weights are highly uneven, then only a small number of samples contribute appreciably to the normalized average, and the estimator becomes correspondingly harder. The quantity $C(\mathbf{x})$ measures exactly this weight concentration. It equals 1 when the weights are constant, and becomes larger when the tilted distribution is harder to approximate using samples drawn from the base distribution $p_x$. For more related discussion see (Aminian et al., 2025).

# G. Going beyond independent inference time sampling: self-consistency and beam search

In this appendix we briefly sketch two speculative extensions of the inference-time sampling-and-selection framework studied in the main text. Both ideas introduce dependence between candidates, going beyond the i.i.d. sampling assumption that enables the clean proportional-limit analysis in this paper. While we do not pursue these directions here, they appear promising and may admit tractable approximations closely related to the calculations already developed.

## G.1. Self-consistency

A common heuristic in LLM decoding is to prefer answers that are not only high-reward individually, but also supported by other sampled candidates. This idea is often referred to as self-consistency: candidates that agree with many others are treated as more reliable. We can model this by augmenting the selection score with a consensus kernel:

$$s_i = \sum_{j \neq i} K_\tau(y_i, y_j), \qquad K_\tau(y, y') = \exp\left(-\frac{(y - y')^2}{\tau}\right),$$

and scoring each candidate with

$$\underbrace{r(y_i, \mathbf{x})}_{\text{alignment to reward}} + \beta \underbrace{\log s_i}_{\text{alignment to consensus}},$$

for some $\beta \geq 0$. The softmax then uses $\exp\left([\, r_R(y_i, \mathbf{x}) + \beta \log s_i\,]/T\right)$. For the purpose of analytical calculations it is reasonable to approximate $s_i \to \mathbb{E}[s_i \mid y_i]$ in this expression. This in effect rescales $T$ and shifts $\mathbf{w}_R$. The theoretical analysis can be performed following similar steps as in the paper.

## G.2. Beam search

Another practically important decoding primitive is beam search, which introduces a structured pruning step before selection. We model a simplified, one-step abstraction of this mechanism as follows. Fix a search budget $K$ (total proposals explored by the decoder) and a beam width $B \in \{1, \ldots, K\}$ (candidates kept after pruning). A beam search operator $\mathcal{B}_{K,B}$ keeps the $B$ proposals:

$$\begin{aligned} \{Y_{(1)}, \ldots, Y_{(B)}\} &= \mathcal{B}_{K,B}\left(\{Y_i\}_{i=1}^K\right) \\ &= \arg\text{top-}B \left\{-|Y_i - m|\right\}_{i=1}^K. \end{aligned} \tag{101}$$

These $B$ kept candidates are then passed to our reward-weighted selector (softmax over $r(y, \mathbf{x})$ at temperature $T$).

The marginal distribution of a kept value under $\mathcal{B}_{K,B}$ is well-approximated by (we take correlations into account later):

$$q_{\text{beam}}(y \mid \mathbf{x}, \mathcal{D}) \propto \mathcal{N}(y \mid m, s^2)\, \mathbf{1}\{|y - m| \leq t_B\},$$

where the truncation threshold $t_B$ is chosen so that the expected keep rate matches $B/K$:

$$\Pr\left(|Y - m| \leq t_B\right) = \frac{B}{K}$$

Under symmetric truncation, the mean remains $m$ and the variance $s^2$ contracts to $s_{\text{beam}}^2$.

Even if $Y_1, \ldots, Y_K$ are i.i.d., the kept vector $(Y_{(1)}, \ldots, Y_{(B)})$ is not independent (order-statistic coupling). A standard correction is to replace $k$ by an effective size $k_{\text{eff}}$ such that $Var(\bar{g}_B) = Var(g(Y_{(i)}))/k_{\text{eff}}$, $\bar{g}_B = \frac{1}{B} \sum_{i=1}^B g(Y_{(i)})$. Simplest case would correspond to choosing $g(x) = x$.

In summary, we expect that the replacements $s^2 \to s_{\text{beam}}^2$, $k \to k_{\text{eff}}$ in our formula would capture the effect of beam search.

# H. Experimental details

## H.1. Bayesian linear regression based model

In this Appendix we present additional numerical results for a broader domain of parameters compared to the main text for Bayesian linear regression based model. We see that the patterns explained in the main body of the paper is realized in a broad domain of parameters.

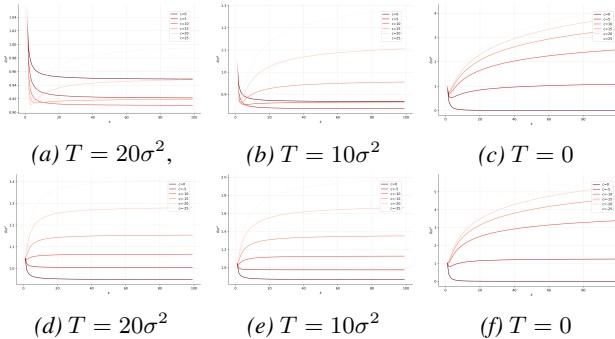

*(a) $T = 20\sigma^2$,*     *(b) $T = 10\sigma^2$*     *(c) $T = 0$*

*(d) $T = 20\sigma^2$*     *(e) $T = 10\sigma^2$*     *(f) $T = 0$*

*Figure 10.* In the plot we have chosen $S = 1, \sigma = 10^{-4}, \gamma = 10^{-3}, n = 10^4, d = 10^1$ and sampled teacher weight $\mathbf{w}_T \sim \mathcal{N}(0, 2^2\mathbf{I})$. We have parameterized the reward weight as follows: $\mathbf{w}_R = (1 + cR/(R + S^2))\mathbf{w}_T$. The plot shows asymmetry between $c > 0, c < 0$ regions as explained in the main text.

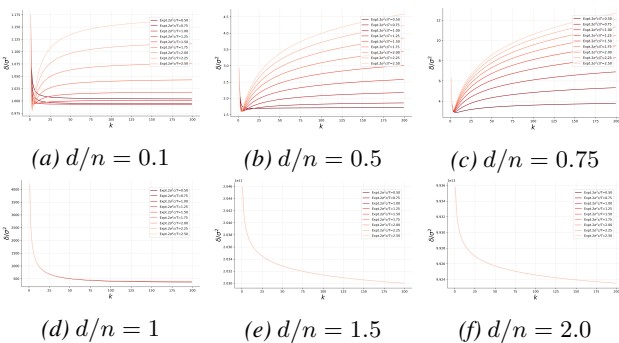

*(a) $d/n = 0.1$*     *(b) $d/n = 0.5$*     *(c) $d/n = 0.75$*

*(d) $d/n = 1$*     *(e) $d/n = 1.5$*     *(f) $d/n = 2.0$*

*Figure 11.* In the plot we have chosen $S = 1, \sigma = 10^{-4}, \gamma = 10^1, n = 10^2, T = 20\sigma^2$ and sampled teacher weight $\mathbf{w}_T \sim \mathcal{N}(0, 2^2\mathbf{I})$. We have parameterized the reward weight as follows: $\mathbf{w}_R = (1 + cR/(R + S^2))\mathbf{w}_T$. We see that for $d \geq n$ generalization error shows different pattern compared to $d < n$. For $d < n$ we see features that are discussed in the main text. We note that as $d/n$ increases $\delta$ at fixed $k$ generally increases. Nevertheless, even for $d \geq n$, an increase in $k$ decreases $\delta$ for a wide range of $\mathbf{w}_R$.

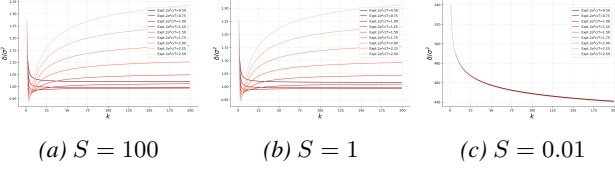

*(a) $S = 100$*     *(b) $S = 1$*     *(c) $S = 0.01$*

*Figure 13.* In the plot we have chosen $T = 20\sigma^2, \sigma = 10^{-4}, \gamma = 10^{-2}, n = 10^2, d = 10^1$ and sampled teacher weight $\mathbf{w}_T \sim \mathcal{N}(0, 2^2\mathbf{I})$. We have parameterized the reward weight as follows: $\mathbf{w}_R = (1 + cR/(R + S^2))\mathbf{w}_T$. The plot shows as $S$ is lowered beyond a critical value we see a sharp change of features.

## H.2. Large language model

Here we present experimental details for LLM based evaluations presented in the main text along with additional

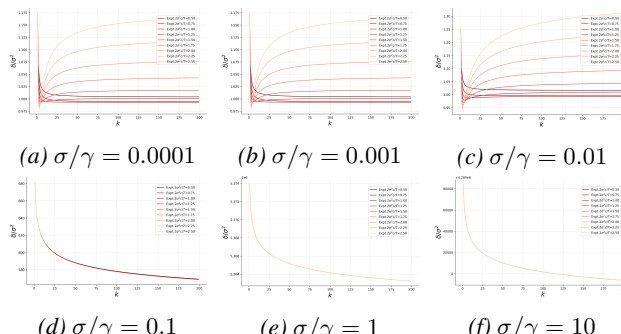

*(a) $\sigma/\gamma = 0.0001$*     *(b) $\sigma/\gamma = 0.001$*     *(c) $\sigma/\gamma = 0.01$*

*(d) $\sigma/\gamma = 0.1$*     *(e) $\sigma/\gamma = 1$*     *(f) $\sigma/\gamma = 10$*

*Figure 12.* In the plot we have chosen $S = 1, \sigma = 10^{-4}, n = 10^2, d = 10^1, T = 20\sigma^2$ and sampled teacher weight $\mathbf{w}_T \sim \mathcal{N}(0, 2^2\mathbf{I})$. We have parameterized the reward weight as follows: $\mathbf{w}_R = (1 + cR/(R + S^2))\mathbf{w}_T$. We see that for large $\sigma/\gamma$ generalization error shows different pattern compared to small $\sigma/\gamma$. Plots show similarity with the plot of $\delta$ vs $d/n$ - in the language of deterministic equivalence both of these are related to the similar change of the un-renormalized ridge.

results.

### H.2.1. Task, prompting, and generation protocol

We evaluate inference-time scaling on GSM8K validation dataset using the `lm-evaluation-harness` task `gsm8k_cot_self_consistency`. Our prompt format is chain-of-thought few-shot prompting with 8 CoT exemplars (8-shot CoT) For each test question, the base model is asked to produce a full reasoning trace followed by a final answer in the format "The answer is {number}". We sample stochastically at generation temperature $0.2$. and extract the final numeric answer using a regular expression matching "The answer is (number)". We also log all reasoning steps to enable Judge-LLM-based selection.

### H.2.2. Base models and judge models

**Base LLMs.** We run inference experiments with a diverse set of base large language models: `Qwen3-8B`, `Llama-3-8B-Instruct`, `Mistral-7B-Instruct-v0.3`, `deepseek-math-7b-rl`, `gemma-3-4b-it`, and `DeepSeek-R1-Distill-Qwen-1.5B`. Inference is executed via the `vLLM` backend through `lm_eval` with model-specific `--model_args`, e.g. tensor/data-parallel settings.

**Judge LLMs.** We score each sampled CoT using two types of judge models: process reward model `Qwen2.5-Math-PRM-7B` and casual LLM `Mistral-7B-Instruct-v0.3`. For the causal judge, we prompt the judge to output a scalar score in $[0, 10]$. We use a short/weak prompt and a detailed/strong prompt. Ex-

amples of each of these prompts are given below. For the PRM, we convert the sampled reasoning into a chat-formatted transcript and insert a step separator token `<extra_0>` after each reasoning step; the PRM score is computed from the model's per-token probability that each separator is classified as "positive" and then scaled to $[0, 10]$. For judges we use deterministic sampling. PRM based judge is known to be much stronger judge compared to the causal model based judge. Our experimental results reflect this - for instance Llama model's accuracy increased from $77\%$ to $87\%$ under PRM judge, whereas under causal judge it went up only by $1\%$.

---

**Model Response Example**

**Weak prompt:** Score 0-10:
[Q] Janet's ducks lay 16 eggs per day. She eats three for breakfast every morning and bakes muffins for her friends every day with four. She sells the remainder at the farmers' market daily for $ 2 per fresh duck egg. How much in dollars does she make every day at the farmers' market?
[A] Janet's ducks lay 16 eggs per day. She eats 3 and bakes 4, so she has 16 - 3 - 4 = 9 eggs left. She sells them for 2 dollars each, so she makes 9 x 2 = 20 dollars. The answer is 18.
[Final] 18
[Score]:
**Judge's response: [0]**

---

**Model Response Example**

**Strong prompt:** Score the answer of the student based on its logical consistency of reasoning and accuracy of the final answer. The output should be a number between 0 to 10.
[question] Janet's ducks lay 16 eggs per day. She eats three for breakfast every morning and bakes muffins for her friends every day with four. She sells the remainder at the farmers' market daily for $ 2 per fresh duck egg. How much in dollars does she make every day at the farmers' market?
[reasoning of the student] Janet's ducks lay 16 eggs per day. She eats 3 and bakes 4, so she has 16 - 3 - 4 = 9 eggs left. She sells them for 2 dollars each, so she makes 9 x 2 = 20 dollars. The answer is 18.
[final answer of the student] 18
[Score]:
**Judge's response: [5]**

---

### H.2.3. ADDITIONAL RESULTS

In this section we present some more empirical results. For the causal judge we use the strong prompt and the for PRM

we process the response into multiple steps - we break on sentences that contains an equal sign.

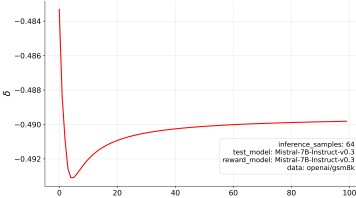

*Figure 14.* Mistral-7B-Instruct-v0.3 as the base model

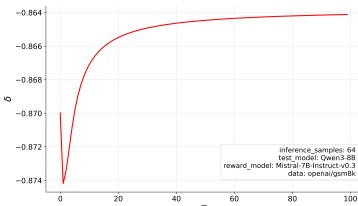

*Figure 15.* Qwen3-8B as the base model

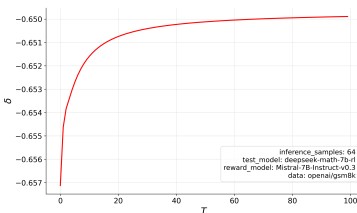

*Figure 16.* deepseek-math-7b-rl as the base model

In the theoretical model we have seen existence of an optimal temperature for the rewarding process. Graphs above present more experimental evidence in support of it.

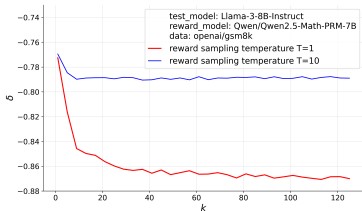

*Figure 17.* Llama-3-8B-Instruct as base model

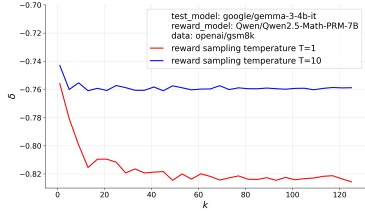

*Figure 18.* gemma-3-4b-it as base model

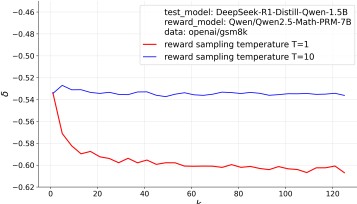

*Figure 19.* DeepSeek-R1-Distill-Qwen-1.5B as base model

These graphs show that for PRM based judge error keeps on decreasing with increase in number of inference time samples almost monotonically at small temperature.

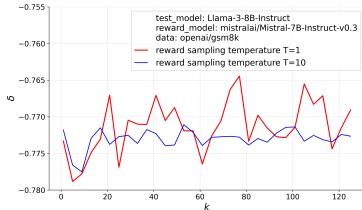

*Figure 20.* Llama-3-8B-Instruct as base model

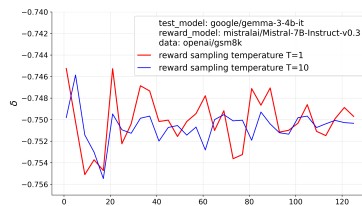

*Figure 21.* gemma-3-4b-it as base model

These graphs show that for causal LM based judge there exists an optimal value of the inference time samples. This phenomena is theoretically analyzed in the main text of the paper.

