# OpenReview forum: "Demystifying LLM-as-a-Judge: Analytically Tractable Model for Inference-Time Scaling"
_ICML.cc/2026/Conference — ICML 2026 regular_

### Official Review · Reviewer_8vc3 · 2026-03-03

**Soundness:** 2
**Presentation:** 3
**Significance:** 3
**Originality:** 2
**Overall Recommendation:** 4
**Confidence:** 3

**Summary:**

The research's core contribution consists of an analytically tractable proxy model—Bayesian linear regression in a teacher–student setup with a quadratic reward and reward-weighted sampling via a softmax temperature—used to derive closed-form/deterministic-equivalent expressions for the posterior predictive mean/variance and to analyze generalization error as a function of k, reward misspecification (w_R vs. w_T), and temperature  T . The paper claims (i) monotone improvement with  k  when reward is sufficiently aligned with the teacher, (ii) non-monotone behavior and a finite optimal  k  under substantial reward misspecification, (iii) an optimal reward temperature at fixed  k , and (iv) in the best-of- k /T=0 limit with teacher reward, error decays as Θ(1/ k ^2) with a leading coefficient derived via extreme value theory.

**Compliance With Llm Reviewing Policy:**

Affirmed.

**Final Justification:**

I appreciate the authors' rebuttal for clarifying some questions and limitations.  However, the theory proposed in this paper has a relatively narrow scope of application. Some conclusions rely on strong assumptions, and its connection to the actual experiments is somewhat weak.

**Key Questions For Authors:**

- Result 1 states validity for “sufficiently small noise” with an implicit threshold σ_c(α, R): what exactly are the required asymptotic assumptions (on σ, γ, Σ spectrum, and k growth with d,n), and how sensitive are the conclusions to violating them? If the regime is narrow, my soundness/significance would drop.
- For the claim that the reward optimizing inference-time selection generally differs from the teacher (w_R ≠ w_T optimal for fixed (k,T)), can you provide the explicit optimizer characterization (closed form or necessary conditions) and an intuitive explanation of when/why it departs from w_T? A clear characterization would strengthen originality and interpretability; otherwise it risks being anecdotal.

**Limitations:**

no
1. Add an explicit “Limitations” section discussing (a) linear-Gaussian proxy mismatch to LLM generation, (b) dependence on reward model calibration/misspecification, and (c) when deterministic-equivalent predictions may fail (e.g., heavy-tailed features, non-isotropic Σ).
2. Discuss potential negative impacts: over-reliance on imperfect judges can amplify systematic biases or adversarial reward hacking; clarify mitigation and safe deployment considerations.

**Strengths And Weaknesses:**

Soundness: The modeling choices (Bayesian linear regression, teacher–student data, quadratic reward, softmax reweighting) are clearly specified, and Results 1–2 align with known Bayesian/regression and deterministic-equivalent techniques; however, key conditions (“sufficiently small noise,” proportional limit derivations, and the Θ(1/k^2) claim) are only summarized with proof sketches here.

Presentation: The setup (training posterior predictive, inference-time algorithm, reward definition, and error metric) is relatively clear in Sections 2–3.

Significance: Inference-time compute allocation and LLM-as-a-judge selection are practically important and understudied theoretically; a solvable model that predicts regimes where more samples hurt and where temperature is optimal could influence how practitioners tune best-of-k/reranking under imperfect judges, even if the proxy is simplified.

Originality: The work’s novelty appears to be the specific combination of (i) reward-weighted sampling as an inference-time selection mechanism, (ii) high-dimensional deterministic-equivalent analysis of generalization error under reward misspecification, and (iii) a best-of-k extreme-value scaling characterization; the excerpt references related scaling-law/voting theory but does not yet precisely position the closest analytical competitors.

---

> ### Author Rebuttal · Authors · 2026-03-27
>
> We thank the reviewer for careful reading of our paper.
> We address each of the concerns below.
>
> **1. Exact asymptotic assumptions.**
> We agree to clarify the meaning of sufficiently small noise in the theorem itself in a revised version of our paper. The intended scope of the theorem  is as follows:
>
> - the proportional limit $ d,n\to\infty$  with $ \alpha=d/n\in(0,1)$  fixed,
> - a covariance matrix $ \bf{\Sigma}$  whose average variance is $ O(1)$  per coordinate, equivalently $ \operatorname{Tr}(\bf{\Sigma})$$=\Theta(d)$ ,
> - the deterministic-equivalent step is taken first, with $ k$  and $ T$  treated as fixed parameters. Next $ T$  is taken large or small as subsequent limits after substituting the deterministic-equivalent predictive distribution.
>
>
> The reason for the small-noise condition is that the predictive mean contains an additional label-noise contribution $ Z(x)$  whose variance, in the proportional limit, obeys (as explained in Appendix B)
> $$
> \operatorname{Var}(Z(x))\to \sigma^2\,\frac{\alpha m_{\Sigma}^{(2)}(R)}{1-\alpha m_{\Sigma}^{(2)}(R)},
> \qquad
> m_{\Sigma}^{(2)}(R):=\frac1d\operatorname{Tr} \big[\Sigma^2(\Sigma+RI)^{-2}\big].
> $$
> Therefore a sufficient condition for dropping this correction is
> $$
> \sigma^2\ll \sigma_c^2(\alpha,R),
> \qquad
> \sigma_c^2(\alpha,R):=\frac{1-\alpha m_{\Sigma}^{(2)}(R)}{\alpha m_{\Sigma}^{(2)}(R)}.
> $$
> In the isotropic case $\bf{\Sigma}$$=S^2I$, this reduces to $ m_{\Sigma}^{(2)}(R)=S^4/(S^2+R)^2$ , in this case for any $\alpha<1$ we can find a non-empty open set for $\sigma$ within which our theorem remains valid. In practice, we perform numerical analysis for $\alpha$ extending from vary small value to close unity in Appendix G.1 Figure 10, 11 and find that predictions of our results remain valid qualitatively until when $\alpha$ is very close to $1$ from below.
>
> **Why ideal reward departs from the teacher.**
>  A short calculation from the high-temperature expansion yields a clean necessary condition for this to be valid as we explain next.
> $$
> \delta(x)=\delta_0(x)-\frac{a}{t(x)}C_1(x)+\frac{b}{t(x)^2}C_2(x)+O \big(t(x)^{-3}\big),
> \qquad
> a=1-\frac1k,
> \quad
> b=\left(1-\frac1k\right)\left(1-\frac2k\right).
> $$
> Using
> $$
> C_1=2\Delta_T\Delta_R+s^2,
> \qquad
> C_2=2\Delta_T\Delta_R+s^2+\Delta_R^2,
> $$
> and differentiating with respect to $ \Delta_R$  gives
> $$
> 0=\partial_{\Delta_R}\delta(x)
> =-\frac{2a}{t(x)}\Delta_T(x)
> +\frac{2b}{t(x)^2}\big(\Delta_T(x)+\Delta_R(x)\big)
> +O \big(t(x)^{-3}\big).
> $$
> Hence, to this order,
> $$
> \Delta_R^{\star}(x)=\left(\frac{a}{b}t(x)-1\right)\Delta_T(x)+O \big(t(x)^{-1}\big)
> =\left(\frac{k}{k-2}t(x)-1\right)\Delta_T(x)+O \big(t(x)^{-1}\big).
> $$
> In the isotropic specialization $ \bf{\Sigma}=S^2I$ , where $ \bf{B_R}=\beta I$  with $ \beta=R/(R+S^2)$ , this implies
> $$
> w_R^{\star}\approx
> \left[1+\beta\left(\frac{k}{k-2}t-2\right)\right]w_T,
> $$
> so the shift is linear in $ T$  to leading order. The current draft states only the dominant large-$ t$  piece of this relation.
>
> The reason the optimum can move away from $ w_T$  is that the posterior predictive mean approaches the the teacher, $ m(x)=A_R\mu_T(x)$  from below in this higher dimensional limit. In other words we know the weakness of our model and how it approaches the correct answer. We can use this knowledge to leverage predictions.  A finite-temperature reranker centered exactly at $ \mu_T$  does not fully compensate that shrinkage. A slightly shifted reward can counteract the predictor's bias, so the selection optimum need not coincide with the data-generating optimum. We will add this discussion in a revised version more clearly.
>
> **3. Limitations and negative impacts.**
> We agree and will add a dedicated section. The section will therefore state clearly:
>
> - what is included (general covariance structure under the Gaussian proxy, note that non-isotropic $ \bf{\Sigma}$  is actually already covered by our main theorem),
> - what is not included (heavy tails, non-Gaussian covariates, token-level LLM generations),
> - the safety implication that stronger selection can amplify judge errors leading to reward hacking. For mitigation we will explicitly recommend validation-based tuning of $ k$  and $ T$  leading to judge calibration.

---

> > ### Author Rebuttal · Reviewer_8vc3 · 2026-04-03
> >
> > I appreciate the mathematical justification using the Taylor expansion for the low-temperature regime. However, a Taylor expansion is strictly a local approximation. In tasks like mathematical reasoning (GSM8K), generated answers might cluster near the true optimum. But what about more open-ended or creative tasks where the generated candidate distribution is highly multi-modal and has high variance? In such scenarios, the samples are far from the local optimum, and the quadratic approximation should mathematically break down. Would your theoretical predictions (such as the existence of an optimal $k$) still qualitatively hold in these high-variance, multi-modal generative tasks, or is this theory strictly bounded to convergent reasoning tasks?

---

> > > ### Author Response · Authors · 2026-04-03
> > >
> > > We thank the reviewer for the insightful follow-up questions. In the revision, we will add an appendix dedicated to the general reward model and add a discussion in the limitations section as suggested by the reviewer.
> > >
> > >
> > > **1. High temperature analysis of a generic reward model.**
> > >
> > > The reason we see a non-monotonic nature of generalization error as a function of $k, T$ is related to the relative magnitude of  $r/T$ and $r^2/T^2$ terms in high temperature expansion - this fact does not depend on the specific form of the reward (being quadratic) or the predictive distribution (being Gaussian), as we explain next.
> > >
> > > Let $$p_x(y):=p(y\mid x,\mathcal D)$$ be an arbitrary predictive distribution for a fixed input $x$, let $r(y,x)$ be an arbitrary reward, and let $\ell(y,x)$ be the task loss we care about. Define the finite-sample reranked loss $$\delta_k^{(\ell,r)}(x,T):=\mathbb E_{Y_{1:k}\sim p_x}\left[\frac{\sum_{i=1}^k \ell(Y_i,x)e^{r(Y_i,x)/T}}{\sum_{j=1}^ke^{r(Y_j,x)/T}}\right].$$Expanding in $\varepsilon:=1/T$ gives $e^{\varepsilon r}=1+\varepsilon r+\frac{\varepsilon^2}{2}r^2+O(\varepsilon³).$ Using this yields
> > > $$\delta_k^{(\ell,r)}(x,T)=\mu_\ell(x)+\frac{k-1}{k}\frac{1}{T} Cov_x(\ell,r)+\frac{(k-1)(k-2)}{2k^2}\frac{1}{T^2} Cov_x  \big(\ell,(r-\mu_r(x))^2\big)+O(T^{-3}),$$
> > > where
> > > $\mu_\ell(x):=\mathbb E_{p_x}[\ell(Y,x)],\mu_r(x):=\mathbb E_{p_x}[r(Y,x)],$
> > > and $Cov_x$ denotes covariance under $Y\sim p_x$.
> > >
> > > * First-order term.
> > > To first order, reranking is beneficial precisely when $Cov_x(\ell,r)<0,$ that is, when a larger reward is correlated with a smaller task loss under the base predictive distribution. At this order, the dependence on $k, T$ is monotone. In the asymptotic high-$T$ regime where the first-order term dominates, only the basic alignment effect survives.
> > >
> > >
> > > * Second-order term.
> > >    * If $Cov_x  \big(\ell,(r-\mu_r)^2\big)>0,$ then higher-loss draws tend to occur where the reward fluctuates more strongly around its mean. This is exactly the mechanism that can generate a finite optimal temperature or a finite optimal sample budget.
> > >    * If $Cov_x  \big(\ell,(r-\mu_r)^2\big)<0,$ then lower-loss draws tend to come with larger reward spread. In that case, the second-order correction helps, and reranking remains helpful.
> > >
> > > The expansion above does not use the Gaussianity of the predictive distribution. It only requires the relevant moments to exist under $p_x$. Hence, this can be easily generalized to heavy-tailed or multi-modal distributions. We agree with the reviewer that a detailed study of it is an interesting future direction.
> > >
> > > **2. Low temperature analysis of a generic reward model.**
> > >
> > > Here, we derive a more general scaling law at low temperature for a generic setup that can fit the curves in Figure 9 for LLMs under a mild set of assumptions. We restrict our study to the case when the minima of the reward $y_{\star}(x)$ coincide with the target (if this is not the case, then the generalization error saturates to a constant value at large $k$, please see response to Reviewer XGcN for this case, due to character limitation, we are not able to write it here), let
> > > $$D_k(x):=\min_{1\le i\le k}|Y_i-y_{\star}(x)|, Y_i\stackrel{\mathrm{iid}}{\sim} p_x.$$ Assume the density at the target for a given input $x$ follows a power law $$p_x(y_{\star}+u)= c_x |u|^{\beta}+o(u^{\beta}),$$ as $u\to 0$. We emphasize this condition is different from a heavy-tailed one. Then, $$Pr(|Y-y_{\star}(x)|\le r\mid x)=2c_x\int_0^r u^\beta du=\frac{2c_x}{\beta+1}r^{\beta+1}=A_x r^{\beta+1}.$$
> > > Plugging in the small-$r$ approximation, $$Pr(D_k(x)>r\mid x)\approx\bigl(1-A_x r^{\beta+1}\bigr)^k\approx\exp  \big(-kA_x r^{\beta+1}\big)$$
> > > Using integration by parts
> > > $$\mathbb E[D_k(x)^2\mid x]=2\int_0^\infty r \Pr(D_k(x)>r\mid x) dr.$$
> > > Substitute the asymptotic form above and change variables to get
> > > $$\mathbb E[D_k(x)^2\mid x]\approx2\int_0^\infty re^{-kA_x r^{\beta+1}}dr\approx2k^{-2/(\beta+1)}\int_0^\infty te^{-A_x t^{\beta+1}}dt.$$
> > > That is $$\mathbb E[D_k(x)^2\mid x]=\Theta(k^{-2/(\beta+1)}).$$ For Gaussian distribution in the paper $\beta=0$ and $c_x=f_x(\mu_T(x))$, this gives $1/k^2$ scaling of the generalization error.  Figure 9 for LLMs can fit in well with $\beta>0$. For heavy-tailed or multi-modal distributions, only the coefficient $\beta$ is important for the scaling law. Note that our scaling law is not about the success rate in one of $k$ trials as is usual in other studies, but rather our error takes the reasoning of every solution takes into account.
> > >
> > > **Summary.**
> > >
> > > We acknowledge that if the density of the predictive distribution is non-smooth near the target or lacks proper convergence properties of its moments, our formalism would require significant modifications. We will add these in the limitation section following the reviewer's suggestions. If you find our explanation satisfactory and believe this is an important theoretical work, please consider increasing the score. Thank you so much for the review.

---

### Official Review · Reviewer_5owz · 2026-03-10

**Soundness:** 3
**Presentation:** 3
**Significance:** 4
**Originality:** 3
**Overall Recommendation:** 4
**Confidence:** 3

**Summary:**

This paper provides a theoretical analysis of test-time scaling, focusing on the roles of temperature, sampling number k, and dataset size n. In particular, the authors analyze why best-of-k and explain the existence of optimal sampling temperature and optimal sampling number k.

**Compliance With Llm Reviewing Policy:**

Affirmed.

**Final Justification:**

This paper focuses on a timely problem with solid theoretical analysis. Despite that some settings are simple, useful and interesting insights are provided. Due to these reasons, I maintain my score for weak acceptance.

**Key Questions For Authors:**

- Can the authors extend their theoretical results beyond quadratic reward functions to more general reward formulations?
- Do the main conclusions still hold under non-i.i.d. data distributions?
- Can the authors clarify whether the linear generative assumption conflicts with empirical observations in real large language models?

**Limitations:**

The authors should explicitly discuss the limitations related to the simplified reward modeling and the linear generative assumption raised above.

**Strengths And Weaknesses:**

Strengthes:
- The paper studies a highly important and timely problem: why inference-time scaling works and why best-of-k can sometimes degrade performance.
- The authors provide a rigorous theoretical analysis under a Bayesian framework.
- The paper presents interesting and valuable insights, particularly that when the reward is misaligned with the true objective, best-of-k may become harmful, and in the optimal regime the reward direction does not necessarily coincide with the true objective.

Weaknesses:
- The reward model is overly simplified, as it only considers a quadratic reward. It is unclear whether this formulation can realistically approximate practical reward models, and whether the theoretical results extend to more general and commonly used reward functions. The current manuscript lacks a broader generalization of the theory.
- The authors assume that the data are i.i.d., which may be overly idealized. In many real-world scenarios, data distributions are long-tailed or non-i.i.d.
- The generative model (the mapping from x to y) is assumed to be linear, which is highly simplified. Modern large language models are strongly nonlinear, so this assumption may limit the applicability of the conclusions.

---

> ### Author Rebuttal · Authors · 2026-03-27
>
> We thank the reviewer for careful reading of our paper and the positive assessment.
> We address each of the concerns below.
>
>
> **1. Beyond quadratic reward functions.**
> The quadratic reward can be generalized as follows:
> *Low-temperature view.* If a judge-induced score is a smooth scalar function $ r(y,x)$  and it has a local maximizer $ y_\star(x)$ , then a second-order Taylor expansion gives
> $$
>  r(y,x)=r(y_\star,x)-\frac{\kappa(x)}{2}\,(y-y_\star(x))^2 + O \left(|y-y_\star(x)|^3\right),\qquad \kappa(x):=-\partial_y^2 r(y_\star,x)>0.
> $$
> Thus, locally, the reward-weighted selector behaves like our quadratic model with
> $$
> \mu_R(x)\longleftrightarrow y_\star(x).
> $$
> Note that $r(y_\star,x)$ drops out while taking the softmax. When $T$ is small, only contributions that are very close to the $ y_\star(x)$ contribute significantly to the generalization error, hence in this regime discussion of quadratic reward approximation is particularly suitable without loss of generality.
>
> *High-temperature view.* When $ T$  is large, the reweighting enters through the expansion
> $$
>  e^{r(y,x)/T}=1+\frac{r(y,x)}{T}+\frac{r(y,x)^2}{2T^2}+\cdots,
> $$
> So, the first corrections depend only on low-order reward moments under the predictive distribution. In the high temperature regime, our calculations can be easily further generalized order by order in $1/T$ perturbation theory beyond quadratic order. In revision, we will add a paragraph explaining that the quadratic reward is the universal second-order local model of a smooth verifier, while also stating clearly that a full general-reward theorem is left to future work.
>
>
>
> **2. On non-i.i.d sampling.**
>
> We discuss an extension of our results to non i.i.d. sampling - self-consistency and beam search in appendix F. We carefully discuss set of approximations that will allow us to study these set ups in details. In particular for beam search we show certain replacements  $s^2 \to s_{beam}^2, k\to k_{eff}$ of our formula will capture the effect of correlated sampling as we explain below.
> Fix a search budget $ K$  (total proposals explored by the decoder) and a beam width $ B\in\{1,\dots,K\}$  (candidates kept after pruning).
> A beam search operator $ O_{K,B}$  keeps the $ B$  proposals:
> $${Y_{(1)},\dots,Y_{(B)}}=O_{K,B} (Y_i: i=1,..,K)=\arg\mathrm{top}\text{-}B (-|Y_i-m|: i=1,.., K).$$
> These $ B$  kept candidates are then passed to our reward-weighted selector (softmax over $ r(y,x)$  at temperature $ T$ ).
>
>
> The marginal distribution of a kept value under $ O_{K,B}$  is well-approximated by (we take correlations into account later):
> $$q_{\mathrm{beam}}(y \mid x, D) \propto \mathcal{N}(y\mid m,s^2)\mathbf{1} (|y-m|\le t_B),$$
> where the truncation threshold $ t_B$  is chosen so that the expected keep rate matches $ B/K$ :
> $$
> \text{Pr} \big(|Y-m|\le t_B\big)\;=\;\frac{B}{K}
> $$
> Under symmetric truncation, the mean remains $ m$  and the variance $s^2$ contracts to $s_{\mathrm{beam}}^2$.
>
> Even if $ Y_1,\dots,Y_K$  are i.i.d., the kept vector $ (Y_{(1)},\dots,Y_{(B)})$  is not independent (order-statistic coupling). A standard correction is to replace  $ k$  by an effective size $k_{\mathrm{eff}}$ such that $Var(\bar{g})=$ $\frac{Var(g(Y_{(i)})}{k_{\mathrm{eff}}}, \quad \bar{g}=\tfrac{1}{B}\sum_{i=1}^B g(Y_{(i)}).$ Simplest choice that captures leading order effect is $g(x)=x$. This gives a clear method for choosing  $s^2 \to s_{beam}^2, k\to k_{eff}$  in our formula.
>
>
>
>
>
>
>
> **3. On the linear teacher assumption vs real LLMs.**
> We agree that the model is not a literal model of token generation in an LLM. To the best of our knowledge so far empirical results in the linear model does match with that of LLM qualitatively not quantitatively. We agree that this correspondence needs to be stated much more explicitly:
> - the base LLM's sampled responses play the role of draws from a predictive distribution of the regression model,
> - the judge LLM score plays the role of reward,
> - the reranking softmax temperature is the analogue of the same in the theory,
> - weak vs strong judges are the empirical proxy for changing reward quality,
> - the LLM experiments are intended to validate qualitative predictions of the theory:  monotone vs non-monotone behavior as inference time sample grows depending on judge quality: existence of an optimal number of inference time samples (Figure 4 for theory, Figure 7 experiment) and best-of-$k$ power-law-like decay under a strong judge (Figure 6 for theory, Figure 9 experiment), existence of an optimal reranking temperature (Figure 5 for theory, Figure 7 experiment).
>
>
>
>
>
> **4. Limitations.**
> We agree and will add an explicit limitations section covering exactly the three issues you raise: the quadratic reward proxy, the scope of i.i.d. sampling, and the fact that the LLM experiments are qualitative validations rather than literal parameter recovery from the theory.

---

> > ### Author Rebuttal · Reviewer_5owz · 2026-04-01
> >
> > The clarification on the quadratic reward function addresses my previous concern. In particular, the low-temperature regime is well justified via a local quadratic approximation. However, in the high-temperature regime, although the quadratic term still exists, the behavior appears to be dominated by the linear (first-order) term. It would be helpful if the authors could further analyze whether the main results still hold in this regime, or clarify their dependence on the quadratic structure. Including such discussion would strengthen the paper.
> >
> > The clarification regarding the extension from i.i.d. to non-i.i.d. settings is also appreciated.
> >
> > In general, I find the correspondence between the theoretical variables and LLM components to be reasonable. The experimental results exhibit trends consistent with the theoretical predictions, and I believe the qualitative insights provided by the paper are valid. However, most theoretical results rely on the linear teacher assumption, which implies a Gaussian predictive distribution. In more general nonlinear settings, this assumption may not hold, and the theoretical guarantees may not directly extend. A brief discussion of this limitation would further strengthen the paper.
> >
> > Follow-up questions:
> > 1. When the linear (first-order) term dominates in the high-temperature regime, do the main theoretical results (e.g., optimal k temperature behavior) still hold, or do they qualitatively change? If not, can the authors characterize the differences?
> > 2. To better understand the scope of the theory, how would its applicability change if the predictive distribution deviates from Gaussian (e.g., becomes multi-modal or heavy-tailed)? It would be helpful if the authors could discuss whether such deviations introduce additional gaps between theory and practice.

---

> > > ### Author Response · Authors · 2026-04-02
> > >
> > > We thank the reviewer for the very thoughtful follow-up. In the revision, we will add an appendix dedicated to the general reward model and add a discussion in the limitations section as suggested by the reviewer.
> > >
> > >
> > > **1. Role of the linear vs quadratic terms of reward in generalization error.**
> > >
> > > Before we discuss mathematical details, note that in Figure (2) of our paper we compares two situations (a) left graph, at higher temperature compared to (b) right graph. One of the key difference between these two graphs is the following: the range of reward mis-specifications that leads to monotonic decay of generalization error is larger for high temperature graph (a). This is related to the fact that at higher temperature the effect of $r/T$ term dominates over $r^2/T^2$ term more for a given mis-alignement level of the reward model as the reviewer pointed out. In other words, the reason we see non-monotonic nature as a function of $k, T$ is related to the relative magnitude of  $r/T$ and $r^2/T^2$ terms - this fact does not depend on the specific form of the reward (being quadratic) or the predictive distribution (being Gaussian) as we explain next.
> > >
> > > Let $$p_x(y):=p(y\mid x,\mathcal D)$$ be an arbitrary predictive distribution for a fixed input $x$, let $r(y,x)$ be an arbitrary reward, and let $\ell(y,x)$ be the task loss we care about. Define the finite-sample reranked loss $$\delta_k^{(\ell,r)}(x,T):=\mathbb E_{Y_{1:k}\sim p_x}\left[\frac{\sum_{i=1}^k \ell(Y_i,x)e^{r(Y_i,x)/T}}{\sum_{j=1}^ke^{r(Y_j,x)/T}}\right].$$Expanding in $\varepsilon:=1/T$ gives $e^{\varepsilon r}=1+\varepsilon r+\frac{\varepsilon^2}{2}r^2+O(\varepsilon³).$ Using this yields
> > > $$\delta_k^{(\ell,r)}(x,T)=\mu_\ell(x)+\frac{k-1}{k}\frac{1}{T} Cov_x(\ell,r)+\frac{(k-1)(k-2)}{2k^2}\frac{1}{T^2} Cov_x  \big(\ell,(r-\mu_r(x))^2\big)+O(T^{-3}),$$
> > > where
> > > $\mu_\ell(x):=\mathbb E_{p_x}[\ell(Y,x)],\mu_r(x):=\mathbb E_{p_x}[r(Y,x)],$
> > > and $Cov_x$ denotes covariance under $Y\sim p_x$.
> > >
> > > * First-order term.
> > > To first order, reranking is beneficial precisely when $Cov_x(\ell,r)<0,$ that is, when larger reward is correlated with smaller task loss under the base predictive distribution. At this order, the dependence on $k,T$ is monotone. So the reviewer is exactly correct that, in the asymptotic high-$T$ regime where the first-order term dominates, only the basic alignment effect survives.
> > >
> > >
> > > * Second-order term.
> > >    * If $Cov_x  \big(\ell,(r-\mu_r)^2\big)>0,$ then higher-loss draws tend to occur where the reward fluctuates more strongly around its mean. This is exactly the mechanism that can generate a finite optimal temperature or a finite optimal sample budget.
> > >    * If $Cov_x  \big(\ell,(r-\mu_r)^2\big)<0,$ then lower-loss draws tend to come with larger reward spread. In that case, the second-order correction helps, and reranking remains helpful.
> > >
> > > **2. Do the main conclusions still hold if the predictive distribution is not Gaussian?**
> > >
> > > We agree this is an important scope question. Here we derive a more general scaling law at low temperature for a generic setup that can fit the curves in Figure 9 for LLMs.
> > >
> > > * High-temperature regime.
> > > The expansion above does not use Gaussianity. It only requires the relevant moments to exist under $p_x$. Hence, this can be easily generalized to heavy-tailed or multi-modal distributions.
> > >
> > >  * Low-temperature regime.
> > > Here the key object is not the global shape of the predictive distribution, but its local behavior near the target answer. In this case only behavior of the reward near the target is important.  When the minima of the reward $y_{\star}$ coincides with the target (if this is not the case then the generalization error saturates to a constant at large $k$, please see response to Reviewer XGcN for this case, due to character limitation we are not being able to write it here), let
> > > $$D_k(x):=\min_{1\le i\le k}|Y_i-y_{\star}(x)|, Y_i\stackrel{\mathrm{iid}}{\sim} p_x.$$ Assume the density at the target for a given input $x$ follows a power law $$p_x(y_{\star}+u)= c_x |u|^{\beta}+o(u^{\beta}),$$ as $u\to 0$. Then, $$Pr(|Y-y_{\star}(x)|\le r\mid x)=2c_x\int_0^r u^\beta du=\frac{2c_x}{\beta+1}r^{\beta+1}=A_x r^{\beta+1}.$$
> > > Plugging in the small-$r$ approximation, $$Pr(D_k(x)>r\mid x)\approx\bigl(1-A_x r^{\beta+1}\bigr)^k\approx\exp  \big(-kA_x r^{\beta+1}\big)$$
> > > Using integration by parts
> > > $$\mathbb E[D_k(x)^2\mid x]=2\int_0^\infty r \Pr(D_k(x)>r\mid x) dr.$$
> > > Substitute the asymptotic form above and change variables to get
> > > $$\mathbb E[D_k(x)^2\mid x]\approx2\int_0^\infty re^{-kA_x r^{\beta+1}}dr\approx2k^{-2/(\beta+1)}\int_0^\infty te^{-A_x t^{\beta+1}}dt.$$
> > > That is $$\mathbb E[D_k(x)^2\mid x]=\Theta(k^{-2/(\beta+1)}).$$ For Gaussian distribution in the paper $\beta=0$ and $c_x=f_x(\mu_T(x))$, this gives $1/k^2$ scaling of the generalization error.  Figure 9 for LLMs can fit in well with $\beta>0$. For heavy tailed or multi-modal distributions only the coefficient $\beta$ is important for the scaling law.

---

### Official Review · Reviewer_XGcN · 2026-03-13

**Soundness:** 4
**Presentation:** 4
**Significance:** 3
**Originality:** 3
**Overall Recommendation:** 5
**Confidence:** 4

**Summary:**

This work studies the inference scaling laws with LLM as a judge under the theoretical framework of linear regression.

**Compliance With Llm Reviewing Policy:**

Affirmed.

**Final Justification:**

This paper uses the linear model to understand the inference-time scaling. This is the first work, as far as I know, that uses a **white-box** model to study inference-time scaling. The work analyzes the high- and low-misspecification settings, which are consistent with empirical findings. The white-box model casts more insight into the training behavior and inference performance.

Before the rebuttal, the major drawback was the lack of comparison against existing works on inference-time scaling that study **black-box** models. This is issue is now resolved in the inference, and I am expecting to see these discussions in the revision.

Therefore, I would strongly recommend acceptance.

**Key Questions For Authors:**

1. How do the theoretical results in this paper compare against Soft BoN [1]?
2. How does the property that "the generalization error decreases monotonically with the number of inference-time samples" compare with the scaling monotonicity in [2]?
3. How is the coverage coefficient in previous works (e.g., [3]) related to the theoretical framework in this paper.
4. Is it posible to show the two regimes $\mathcal R$ and $\mathcal R^C$ in the experiments in Section 4?

[1] Aminian et al. Best-of-n through the smoothing lens: Kl divergence and regret analysis. 2025.

[2] Di et al. Best-of-Majority: Minimax-Optimal Strategy for Pass@k Inference Scaling. 2025.

[3] Huang et al. Is best-of-n the best of them? coverage, scaling, and optimality in inference-time alignment. 2025.

**Limitations:**

Yes.

**Strengths And Weaknesses:**

## Strengths

1. It is good to see a number of theoretical results on test-time scaling unified under the proposed theoretical framework.

## Weaknesses

### Soundness

1. $R$ seems undefined in Result 1.

### Presentation

2. The convention is to use $N$ instead of $k$ to denote the number of responses generated, with the well-known algorithm Best-of-N (BoN).
3. Figure 1 could be enlarged.
4. The paragraph before Result 1 is very informative but currently a bit messy. There are several findings: (i) 2 regimes, one ($\mathcal R$) with monotonic accuracy in $k$ and one ($\mathcal R^C$) without; (ii) Temperature $T$ makes a difference, and (iii) in the regime $\mathcal R$, $w_R=w_T$ does not minimize $\delta$. The correspondence of these empirical findings with the theoretical results can also be better presented.

### Significance

5. The theoretical result about the low-temperature regime without accurate reward function is lacking.

### Originality

6. The algorithm studied in this paper looks very close to the Soft BoN [1].
7. The authors could discuss some additional works on test-time scaling [2, 3, 4], especially the connection and differences of the findings in this paper (each subsection title in Section 3) and in previous works.

Overall, I think the major drawbacks of this paper are (i) presentation of Section 3, and (ii) discussion of results compared with prior works. Currently there isn't a single citation of prior works in Section 3.

[1] Aminian et al. Best-of-n through the smoothing lens: Kl divergence and regret analysis. 2025.

[2] Di et al. Best-of-Majority: Minimax-Optimal Strategy for Pass@k Inference Scaling. 2025.

[3] Wu et al. Inference scaling laws: An empirical analysis of compute-optimal inference for problem-solving with language models. 2024.

[4] Huang et al. Sample complexity and representation ability of test-time scaling paradigms. 2025.

---

> ### Author Rebuttal · Authors · 2026-03-27
>
> We thank the reviewer for careful reading of our paper and the comments.
> We address each of the concerns below.
>
> **1. Undefined \(R\) and presentation fixes.**
> In Result 1, the renormalized ridge \(R\) appears in equation (11) before being defined in equation (12). From equation (12) one has to solve for $R$ in terms of $\hat R$. In the revision, we will move the definition of $R$ before it is used.
> We will also enlarge Figure 1, reformat the paragraph before Result 1 into explicit bullets, and add citations to prior work at the start of each subsection in Section 3. We agree that ``Best-of-$N$'' is standard in the LLM literature. We used \(k\) mainly to avoid conflict with the training set size \(n\).
>
> **2. Comparison to SBoN.**
> We agree that the setup studied in our paper is very closely related to Soft BoN.  We will cite the SBoN works prominently and discuss the connection to our work in the related work section.
>
> - SBoN papers study how the induced policy approaches an optimal tilted distribution and quantify trade-offs in KL divergence/regret bounds relative to that target. The finite inference time sample version of their smoothing is closely related to our definition of generalization error.
> - our paper analyzes a Bayesian regression model with reward and identifies a scaling limit of a large number of samples in higher dimensions, where explicit leading order calculations can be performed instead of relying on bounds. Hence, technically, our paper takes a different route compared to SBoN and presents a formula for the optimal value of reward temperature in Remark 4. It is numerically validated in Figure 5.
>
> **3. Comparison with scaling monotonicity in Best-of-Majority.**
> - Best-of-Majority work says one can redesign the inference algorithm so that monotonicity is guaranteed;
> - Our theory characterizes when the standard reranking primitive is monotone, and when it fails.
>
> **4. Continuous analogue of the coverage coefficient.**
>  In our setup, the closest analogue of the discrete coverage coefficient is:
> $$
>  f_T(x):=p\big(y=\mu_T(x)\mid x,\mathcal D\big)
>  =\frac{1}{\sqrt{2\pi} s(x)}\exp \left(-\frac{\Delta_T(x)^2}{2s(x)^2}\right),
> $$
> where $\Delta_T(x)=m(x)-\mu_T(x)$. Then our theorem for teacher-rewarded best-of-$k$ becomes
> $$
>  \delta(x)=\frac{\pi s(x)^2}{k^2}\exp \left(\frac{\Delta_T(x)^2}{s(x)^2}\right)
>  =\frac{1}{2k^2 f_T(x)^2}.
> $$
>
>
> **5. Low-temperature regime with misspecified reward.**
>
>  We will add the following new result in a revised version of the paper. At $ T=0$ , the selected output is
> $$
> Y_{(R)}:=\arg\min_{1\le i\le k}|Y_i-\mu_R(x)|,
> \qquad Y_i\stackrel{\text{iid}}{\sim}f_x=\mathcal N\big(m(x),s(x)^2\big).
> $$
> Define the predictive density at the reward target,
> $$
>  f_R(x):=f_x \big(\mu_R(x)\big)=\frac{1}{\sqrt{2\pi}\,s(x)}
>  \exp \left(-\frac{(m(x)-\mu_R(x))^2}{2s(x)^2}\right).
> $$
> Near the target it is useful to define
> $$
> H_x(r):=\text{Pr} \big(|Y-\mu_R(x)|\le r\mid x\big)=\int_{-r}^{r} g_x(u)\,du,
> \quad g_x(u):=f_x \big(\mu_R(x)+u\big).
> $$
> For small $r=t/k$, Taylor expansion gives $H_x(\frac{t}{k})=2g_x(0)\frac{t}{k}$.
> When all $k$ samples are far away from the reward target, we get
> $$
> \text{Pr}\big( |Y_{(R)}-\mu_R|>\frac{t}{k} \big)=\bigl(1-H_x (t/k)\bigr)^{k-1}=\left(1-2g_x(0)\frac{t}{k})\right)^{k-1}\longrightarrow e^{-2g_x(0)t}.
> $$
> This implies for large $k$,  $k|Y_{(R)}-\mu_R|$ is an exponential random variable given by rate $ 2f_R(x)$
>  and therefore
> $$
> \mathbb{E} \big[(Y_{(R)}-\mu_R)^2\big]=\frac{1}{2k^2 f_R(x)^2}.
> $$
> Define $S_k:=Y_{(R)}-\mu_R(x)$.  Since $S_k=u$ means that one sample lands at $\mu_R(x)+u$ and all remaining $k-1$ samples lie outside the interval
> $
> [\mu_R(x)-|u|,\mu_R(x)+|u|],
> $
> the density of $S_k$ is
> $$
> h_{k,x}(u)=k\,g_x(u)\,\bigl(1-H_x(|u|)\bigr)^{k-1}.
> $$
> This allows us to calculate the expectation value of $S_k$ using Taylor expansion as before with $u=t/k$. To the leading order in large $k$ we get
> $$
> \mathbb E \big[Y_{(R)}-\mu_R(x)\big]=\frac{f_x' \big(\mu_R(x)\big)}{2\,f_R(x)^3}\,\frac{1}{k^2}.
> $$
> Consequently,
> $$
> \delta(x)=\mathbb{E} \big[(Y_{(R)}-\mu_T)^2\big]
> =(\mu_R-\mu_T)^2+(\mu_R(x)-\mu_T(x))
> \frac{f_x' \big(\mu_R(x)\big)}{f_R(x)^3}\,\frac{1}{k^2}+\frac{1}{2k^2 f_R(x)^2}.
> $$
>  We have presented numerical results for $T=0$ misspecified reward in Appendix G.1 Figure 10-(c), (f) : in particular, note the curve for $c=\pm 5$, in this case, one can see that $\delta/\sigma^2$ saturates near $1$ for large $k$, this is precisely the value predicted by the formula above.
>
>
> **6. Can the two regimes $ \mathcal{R}$  and $ \mathcal{R}^{\mathrm c}$  be shown for LLMs?**
>
> We have shown an operational proxy for moving between these regimes by changing judge quality:
>
> - A stronger PRM behaves like moving closer to the near teacher region, where the $ k$ -curve is closer to monotone. See Figure 9.
> - A weaker Causal LLM behaves like moving toward the misspecified regime, where the $ k$ -curve becomes non-monotone. See Figure 7.

---

> > ### Author Rebuttal · Reviewer_XGcN · 2026-04-02
> >
> > Many thanks for the rebuttal! I am very satisfied with the feedback, especially the new result of low-temperature sampling with reward misspecification. I have decided to raise my rating to 4+. I just have a few followup questions before I increase my rating and compose the final justification:
> >
> > **A1 followup.** Since $R$ needs to be solved w.r.t. $\hat R$ according to (12), how is $\hat R$ defined?
> >
> > **A2 followup.** Many thanks for the comparison with SBoN. Would you show more mathematical details about how "the finite inference time sample version" is related to this work?
> >
> > **A4 followup.** Can the results in this paper be connected with results expressed in the decoupling coefficient? (Maybe this question is related to the **A2 followup**.)

---

> > > ### Author Response · Authors · 2026-04-03
> > >
> > > We thank the reviewer very much for the encouraging follow-up questions and for raising our score. We answer them below.  We will add related discussions to a revised version of our paper.
> > >
> > > **A1. Since $R$ is solved from $\hat R$, how is $\hat R$ defined?**
> > >
> > > The definition of  $\hat R$ appears on the left equation and the second equality in Eq. (12). More explicitly:
> > > $$\hat R:=\frac{\sigma^2}{\gamma^2}\alpha=\frac{\sigma^2}{\gamma^2}\frac{d}{n}.$$
> > > We will organize the theorem to make this definition appear first and clarify that $R$ is a function of $\hat R$. The quantity $\hat R$ is the unrenormalized ridge that appears directly when rewriting the Bayesian posterior covariance in terms of the empirical covariance matrix, as explained in Appendix B.
> > >
> > >
> > > **A2. How the finite-inference-time-sample version is related to SBoN.**
> > >
> > >
> > > Fix $x$ and the training set $\mathcal D$, and write $p_x(y):=p(y\mid x,\mathcal D).$ The reward-reweighted exponentially tilted policy is
> > > $$\pi_T(y\mid x,\mathcal D)=\frac{e^{r(y,x)/T}\,p_x(y)}{Z_T(x)}, Z_T(x):=\int e^{r(y',x)/T}p_x(y')dy'.$$ Now consider the finite-sample procedure in our paper: $Y_1,\dots,Y_k \stackrel{\mathrm{iid}}{\sim} p_x$, $q_i=\frac{e^{r(Y_i,x)/T}}{\sum_{j=1}^k e^{r(Y_j,x)/T}}$, $I\sim\mathrm{Categorical} (q_1,\dots,q_k)$, $Y_{\mathrm{out}}:=Y_I.$ Equivalently, given the sampled set $Y_{1:k}$, this induces the empirical soft policy $\widehat\pi_{k,T}(y):=\sum_{i=1}^k q_i\,\delta_{Y_i}(y).$ For any test function $\varphi$, define $$\widehat\pi_{k,T}(\varphi):=\int \varphi(y)\,\widehat\pi_{k,T}(y) dy=\frac{\sum_{i=1}^k \varphi(Y_i)e^{r(Y_i,x)/T}}{\sum_{j=1}^k e^{r(Y_j,x)/T}}.$$
> > >
> > >
> > > As $k\to\infty$, $\widehat\pi_{k,T}(\varphi) \to \pi_T(\varphi)$. If $\mathbb{E}$ denotes expectation over the inference-time samples $Y_{1:k}$, the generalization error is given by, see equation (9) of our paper, $$\delta(x)=\mathbb{E}(\widehat\pi_{k,T}(\ell)), \quad \ell(y):=(y-\mu_T(x))^2$$. So the finite-$k$ rule studied in our paper is a finite sampled approximation to the reward-reweighted exponentially tilted policy, and our theorem computes the leading corrections to generalization error in a closed form under this policy.
> > >
> > >
> > > **A4. Can the results be connected to a decoupling coefficient?**
> > >
> > > We believe the answer is yes, and there are actually two natural continuous analogues, depending on which aspect one wants to emphasize.
> > >
> > >
> > >   *  A reward-loss coupling coefficient controlling expected finite sample policy difference.
> > >
> > > A natural measure of how strongly reward and task are coupled under the base predictive law $p_x$ is
> > > $$\mathcal A(x):=-\mathrm{Cov}_{p_x}(\ell,r).$$
> > >
> > > If $\mathcal A(x)$ is close to zero, reward and task loss are nearly decoupled under the predictive distribution, and reranking has little first-order effect. If $\mathcal A(x)$ is large and positive, then higher reward correlates with lower task loss, and reranking is beneficial.
> > >
> > > Indeed, the large-$T$ expansion in Result 2 of our paper gives (for a more general derivation, please see the reply to Reviewer 5owz)
> > > $$\mathbb{E}(\widehat\pi_{k,T}(\ell))-\pi_{T}(\ell)=\frac{1}{kT} \mathcal A(x)+O(T^{-2}).$$
> > > So the leading gain from reranking is controlled exactly by this covariance-type coupling parameter.
> > >
> > >  In the quadratic-Gaussian model of our paper, we have,
> > > $$\mathcal A(x)=2s(x)^2 C_1(x),$$
> > >
> > >   * A decoupling coefficient controlling random finite sample policy difference.
> > >
> > > If instead one wants the quantity $\widehat\pi_{k,T}(\ell)-\pi_{T}(\ell)$ that measures how hard it is for a finite sample budget $k$ to realize the ideal tilted policy $\pi_T$, the natural continuous object is the second moment of the importance weights:
> > > $$C(x):=\frac{\mathbb E_{Y\sim p_x}  \bigl[(e^{r(Y,x)/T})^2\bigr]}{\Bigl(\mathbb E_{Y\sim p_x}  \bigl[e^{r(Y,x)/T}\bigr]\Bigr)^2}.$$
> > > If the weights are well spread out, then the estimator behaves like an average of roughly $k$ useful samples, and therefore its error has the usual scale $k^{-1/2}$.
> > > By contrast, if the weights are highly uneven, then only a small number of samples contribute appreciably to the normalized average, and the estimator becomes correspondingly harder.
> > > The quantity $C(x)$ measures exactly this weight concentration. It equals $1$ when the weights are constant, and becomes larger when the tilted distribution is harder to approximate using samples drawn from the base distribution $p_x$. Broadly speaking, Theorem 5.2 in SBoN is philosophically related to these ideas.
> > >
> > >
> > >  In our Gaussian-quadratic model, this can be computed in closed form.
> > > $$C(x)=\frac{1+1/t(x)}{\sqrt{1+2/t(x)}}\exp  \left(\frac{\Delta_R(x)^2/s(x)^2}{(t(x)+1)(t(x)+2)}\right).$$

---

### Official Review · Reviewer_4aWa · 2026-03-16

**Soundness:** 2
**Presentation:** 2
**Significance:** 1
**Originality:** 2
**Overall Recommendation:** 3
**Confidence:** 4

**Summary:**

The authors propose a simplified theoretical framework based on Bayesian linear regression with reward-weighted sampling to analyze inference-time strategies. Under the simplified setting, the paper derives several theoretical results, including conditions under which increasing the number of inference-time samples improves generalization, the existence of an optimal number of samples when the reward model is misaligned, and the existence of an optimal reward temperature. Finally, experiments using several LLMs are presented.

**Compliance With Llm Reviewing Policy:**

Affirmed.

**Final Justification:**

I thank the authors for their clarifications. However, as acknowledged in the rebuttal, the connections between the theory and experiments, as well as the practical implications of the theory, are not yet communicated clearly. While the rebuttal helps clarify several points, these improvements would need to be reflected in the paper itself through substantial revision. Therefore, I increase my score from 2 to 3.

**Key Questions For Authors:**

- Could the authors clarify how the experimental setup corresponds to the assumptions of the theoretical model, and which specific theoretical results the experiments are intended to validate?
- What concrete guidance does the theoretical analysis provide for practitioners using LLM-as-a-judge systems in practice?

**Limitations:**

It seems the impact statement is not included in this paper.

**Strengths And Weaknesses:**

The paper attempts to develop a theoretical framework for analyzing inference-time scaling using a tractable teacher–student regression model. In principle, studying such mechanisms in simplified models is a useful direction. However, a few concerns arise:
- While the paper aims to analyze the trade-off between training-time and inference-time compute, it is unclear whether this framing is meaningful in the context of modern LLM usage. In practice, most users rely on pretrained foundation models and do not control the training process. Instead, inference is typically constrained by user-side compute or budget limits. As a result, the more relevant practical question is how to reduce inference cost while maintaining similar performance. The motivation of this work seems less well aligned with the typical deployment setting of LLMs.
- While the reward model in Equation 6 is claimed to model the LLM-as-a-judge setting, this formulation appears overly simplified. LLM-based judges do not behave like deterministic scoring functions of the form assumed in the paper. Modeling the judge as a simple deterministic quadratic reward may not adequately capture the behavior of real LLM-based evaluations. Thus, it is unclear whether conclusions derived from this simplified assumption would translate to practical LLM settings.
- While the paper includes empirical experiments intended to validate the theoretical predictions, it is difficult to evaluate the validity of these results because the connection between the theoretical model and the experimental setup is not clearly articulated. In particular, it remains unclear how closely the experimental conditions match the assumptions used in the theoretical analysis.

Minor issues:
- The font size of the text in the figures are too tiny.
- Please distinguish between \citep and \citet. For example, Line 97 should use \citet instead of \citep.

---

> ### Author Rebuttal · Authors · 2026-03-27
>
> We thank the reviewer for careful reading of our paper and the comments.
> We address each of the concerns below.
>
> **1. On the training-vs-inference time trade-off.**
> We would like to emphasize that the paper contains several important theoretical results. These theoretical findings are supported by numerical simulations within the theoretical framework, and their corresponding effects are also reproduced qualitatively in large language model experiments. We provide a more detailed discussion of this point later in this reply.
>
> More broadly, much of the theoretical work in deep learning follows this paradigm: theoretical analysis identifies qualitative properties of large-scale systems. In this respect, we believe our paper sets a particularly high standard, especially because, to our knowledge, no existing work simultaneously accounts for all of these features mentioned above.
>
> Beyond these contributions, the paper also provides a theoretical analysis of the trade-off between training-time and inference-time computation. We agree with the reviewer that a detailed empirical study of this trade-off would be costly and may currently be feasible only for a limited number of organizations. However, this does not diminish the significance of the contribution. We therefore respectfully ask that the paper be evaluated with full consideration of the scope and depth of its scientific contributions.
>
>
>
> **2. On the quadratic reward as a LLM-judge proxy.**
>  The quadratic reward is the minimal second-order local model of a smooth verifier around the answer that it most prefers, as we explain next.
>
> *(a) Low-temperature view.* If a judge-induced score is a smooth scalar function $ r(y,x)$  and it has a local maximizer $ y_\star(x)$ , then a second-order Taylor expansion gives
> $$
>  r(y,x)=r(y_\star,x)-\frac{\kappa(x)}{2}\,(y-y_\star(x))^2 + O \left(|y-y_\star(x)|^3\right),\qquad \kappa(x):=-\partial_y^2 r(y_\star,x)>0.
> $$
> Thus, locally, the reward-weighted selector behaves like our quadratic model with
> $$
> \mu_R(x)\longleftrightarrow y_\star(x).
> $$
> Note that $r(y_\star,x)$ drops out while taking the softmax. When $T$ is small, only contributions that are very close to the $ y_\star(x)$  contributes significantly to the generalization error, hence in this regime discussion of quadratic reward approximation is particularly suitable without loss of generality.
>
> *(b) High-temperature view.* When $ T$  is large, the reweighting enters through the expansion
> $$
>  e^{r(y,x)/T}=1+\frac{r(y,x)}{T}+\frac{r(y,x)^2}{2T^2}+\cdots,
> $$
> so, the first corrections depend only on low-order reward moments under the predictive distribution. In the high temperature regime our calculations can be easily further generalized order by order in $1/T$ perturbation theory beyond quadratic order. In the revision, we will add this explanation explicitly.
>
> **3. On how the experiments correspond to the theory.**
> We agree that this correspondence needs to be stated much more explicitly:
> - the base LLM's sampled responses play the role of draws from a predictive distribution of the regression model,
> - the judge LLM score plays the role of reward,
> - the reranking softmax temperature is the analogue of the same in the theory,
> - weak vs strong judges are the empirical proxy for changing reward quality,
> - the LLM experiments are intended to validate qualitative predictions of the theory:  monotone vs non-monotone behavior as inference time sample grows depending on judge quality: existence of an optimal number of inference time samples (Figure 4 for theory, Figure 7 experiment) and best-of-$k$ power-law-like decay under a strong judge (Figure 6 for theory, Figure 9 experiment), existence of an optimal reranking temperature (Figure 5 for theory, Figure 7 experiment).
>
>
>
> **4. Concrete practitioner guidance.**
>  The main ones are:
> - Optimize the judge temperature separately from the model generation temperature. Our temperature $ T$  concerns the reranking softmax, not the base model's decoding temperature. Depending on the quality of the judge, a non-zero value of $T$ might be preferred. See Figure 7 for example.
> - Do not assume larger inference time sample count always helps. With an imperfect judge, increasing $ k$  can eventually hurt. For a stronger judge, this optimal value of $ k$  is larger (compare Figure 7 for Causal LLM as judge and Figure 9 for PRM as judge, also see the theoretical plot Figure 4).
> - If performance degrades as inference time samples $ k$  grows, the right response is not  sample more; it is to increase $ T$ .  Figure (5) and Figure 8 (b) show that a weaker judge leads to a larger value of optimal $T$.
>
>
>
> **5. Impact statement, figure readability, and citation style.**
> We agree on all three points and will fix them in the revision:
> - add an explicit limitations and broader-impact section,
> - enlarge figure labels and legends,
> - correct the citep, citet distinction.

---

> > ### Author Rebuttal · Reviewer_4aWa · 2026-04-03
> >
> > I thank the authors for their clarifications. However, as acknowledged in the rebuttal, the connections between the theory and experiments, as well as the practical implications of the theory, are not yet communicated clearly. While the rebuttal helps clarify several points, these improvements would need to be reflected in the paper itself through substantial revision. Therefore, I increase my score from 2 to 3.

---

> > > ### Author Response · Authors · 2026-04-03
> > >
> > > We thank the review a lot for increasing the score. We will make the promised updates on a revised version of the paper.

---

### Decision · Program_Chairs · 2026-04-30

**Decision:**

Accept (regular)

**Comment:**

The primary remaining concern is in the exposition, which one reviewer feels needs to be improved in clarity.  I encourage the authors to do this, but the other reviewers agree that the paper presents a theoretically interesting perspective relevant to current methods.  While some reviewers believe the setting is too simple, some assumptions have to be made in order for analysis to be tractable.  I strongly encourage the authors to follow the reviewers' suggestions regarding a complete comparison to prior work on the theory of inference-time scaling.